# Diffusion-based Time Series Imputation and Forecasting with Structured State Space Models

**Juan Miguel Lopez Alcaraz**                                    *juan.lopez.alcaraz@uol.de*
*Division AI4Health*
*Oldenburg University*

**Nils Strodthoff**                                              *nils.strodthoff@uol.de*
*Division AI4Health*
*Oldenburg University*

**Reviewed on OpenReview:** *https://openreview.net/forum?id=hHiIbk7ApW*

## Abstract

The imputation of missing values represents a significant obstacle for many real-world data analysis pipelines. Here, we focus on time series data and put forward SSSD, an imputation model that relies on two emerging technologies, (conditional) diffusion models as state-of-the-art generative models and structured state space models as internal model architecture, which are particularly suited to capture long-term dependencies in time series data. We demonstrate that SSSD matches or even exceeds state-of-the-art probabilistic imputation and forecasting performance on a broad range of data sets and different missingness scenarios, including the challenging blackout-missing scenarios, where prior approaches failed to provide meaningful results.

## 1 Introduction

Missing input data is a common phenomenon in real-world machine learning applications, which can have many different reasons, ranging from inadequate data entry over equipment failures to file losses. Handling missing input data represents a major challenge for machine learning applications as most algorithms require data without missing values to train. Unfortunately, the imputation quality has a critical impact on downstream tasks, as demonstrated in prior work (Shadbahr et al., 2022), and poor imputations can even introduce bias into the downstream analysis (Zhang et al., 2022), which can potentially call into question the validity of the results achieved in them.

In this work, we focus on time series as a data modality, where missing data is particularly prevalent, for example, due to faulty sensor equipment. We consider a range of different missingness scenarios, see Figure 1 for a visual overview, where the example of faulty sensor equipment former example already suggests that not-at-random missingness scenarios are significant for real-world scenarios. Time series forecasting is naturally contained in this approach as special case of blackout missingness, where the location of the imputation window is at the end of the sequence. We also stress that the most realistic scenario to address imputation as an underspecified problem class is the use of probabilistic imputation methods, which do not provide only a single imputation but instead allow samples of different plausible imputations.

There is a large body of literature on time series imputation, see (Osman et al., 2018) for a review, ranging from statistical methods (Lin & Tsai, 2020) to autoregressive models (Atyabi et al., 2016; Bashir & Wei, 2018). Recently, deep generative models started to emerge as a promising paradigm to model time series imputation of long sequences or time series forecasting problems at long horizons. However, many existing models remain limited to the random missing scenario or show unstable behavior during training. In addition, we demonstrate that state-of-the-art approaches even fail to deliver qualitatively meaningful imputations in blackout missing scenarios on certain data sets.

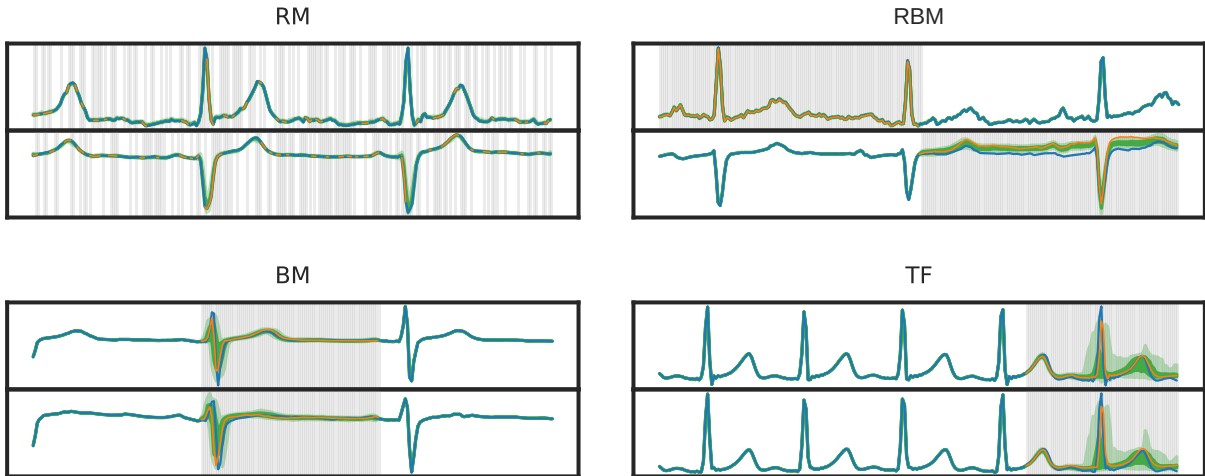

Figure 1: Color scheme introduction. The proposed model SSSD$^{\text{S4}}$ provides imputations for different missingness scenarios that are not only quantitatively but even qualitatively superior, see below, on different data sets for different missingness scenarios (RM: random missing, RBM: random block missing, BM: blackout missing TF: time series forecasting). The signal is in blue, where the white background represents the conditioned ground truth, whereas the gray background represents time steps in specific channels to be imputed. Prediction bands derived from 100 imputations represent quantiles from 0.05 to 0.95 in light green and from 0.25 to 0.75 in dark green. As these bands do not allow visually assessing the quality of individual imputations, we always additionally show a randomly selected single sample in orange.

In this work, we aim to address these shortcomings by proposing a new generative-model-based approach for time series imputation. We use diffusion models as the current state-of-the-art in terms of generative modeling in different data modalities. The second principal component of our approach is the use of structured state-space models (Gu et al., 2022a) instead of dilated convolutions or transformer layers as main computational building block of the model, which are particularly suited to handling long-term-dependencies in time series data.

To summarize, our main contributions are as follows: (1) We put forward a combination of state-space models as ideal building blocks to capture long-term dependencies in time series with (conditional) diffusion models as the current state-of-the-art technology for generative modeling . (2) We suggest modifications to the contemporary diffusion model architecture DiffWave(Kong et al., 2021) to enhance its capability for time series modeling. In addition, we propose a simple yet powerful methodology in which the noise of the diffusion process is introduced just to the regions to be imputed, which turns out to be superior to approaches proposed in the context of image inpainting (Lugmayr et al., 2022). (3) We provide extensive experimental evidence for the superiority of the proposed approach compared to state-of-the-art approaches on different data sets for various missingness approaches, particularly for the most challenging blackout and forecasting scenarios.

## 2 Structured state space diffusion (SSSD) models for time series imputation

### 2.1 Time series imputation

Let $x_0$ be a data sample with shape $\mathbb{R}^{L \times K}$, where $L$ represents the number of time steps, and $K$ represents the number of features or channels. Imputation targets are then typically specified in terms of binary masks that match the shape of the input data, i.e., $m_{\text{imp}} \in \{0, 1\}^{L \times K}$, where the values to be conditioned on are marked by ones and zeros denote values to be imputed. In the case where there are also missing values in the

input, one additionally requires a mask $m_{\mathrm{mvi}}$ of the same shape to distinguish values that are present in the input data (1) from those that are not (0).

One approach of categorizing missingness scenarios in the imputation literature (Rubin, 1976; Lin & Tsai, 2020; Osman et al., 2018) is according to missing completely at random, missing at random and random block missing, where in the first case the missingness pattern does not depend on feature values, in the second case it may depend on observed features and in the last case also on ground truth values of the features to be imputed. Here, we focus on the first category and follow Khayati et al. (2020) for different popular missingness scenarios in the time series domain: *Random missing (RM)*, corresponds to a situation where the zero-entries of an imputation mask are sampled randomly according to a uniform distribution across all channels of the whole input sample. In contrast to Khayati et al. (2020), we consider single time steps for RM instead of blocks of consecutive time steps. This is different for the remaining missingness scenarios considered in this work, where we first partition the sequence into segments of consecutive time steps according to the missingness ration. Firstly, for *random block missing (RBM)* we sample one segment for each channel as imputation target. Secondly, *blackout missing (BM)* we sample a single segment across all channels, i.e., the respective segment is assumed to be missing across all channels, and lastly, *time-series forecasting (TF)* which can be seen as a particular case of BM imputation, where the imputation region spans a consecutive region of $t$ time steps, where $t$ denotes the forecasting horizon across all channels located at the very end of the sequence. The missingness scenarios are described in more detail in Appendix A.3.

## 2.2 Diffusion models

Diffusion models (Sohl-Dickstein et al., 2015) represent a class of generative models that demonstrated state-of-the-art performance on a range of different data modalities, from image (Dhariwal & Nichol, 2021; Ho et al., 2020; 2022a) over speech (Chen et al., 2020; Kong et al., 2021) to video data (Ho et al., 2022b; Yang et al., 2022).

Diffusion models learn a mapping from a latent space to the original signal space by learning to remove noise in a backward process that was added sequentially in a Markovian fashion during a so-called forward process. These two processes, therefore, represent the backbone of the diffusion model. For simplicity, we restrict ourselves to the unconditional case at the beginning of this section and discuss modifications for the conditional case further below. The forward process is parameterized as

$$q(x_1, \ldots, x_T | x_0) = \prod_{t=1}^{T} q(x_t | x_{t-1}) \,, \tag{1}$$

where $q(x_t | x_{t-1}) = \mathcal{N}(\sqrt{1-\beta_t} x_{t-1}, \beta_t \mathbb{1})[x_t]$ and the (fixed or learnable) forward-process variances $\beta_t$ adjust the noise level. Equivalently, $x_t$ can be expressed in closed form as $x_t = \sqrt{\alpha_t} x_0 + (1 - \alpha_t)\epsilon$ for $\epsilon \sim \mathcal{N}(0, \mathbb{1})$, where $\alpha_t = \sum_{i=1}^{t}(1 - \beta_t)$. The backward process is parameterized as

$$p_\theta(x_0, \ldots, x_{t-1} | x_T) = p(x_T) \prod_{t=1}^{T} p_\theta(x_{t-1} | x_t) \tag{2}$$

where $x_T \sim \mathcal{N}(0, \mathbb{1})$. Again, $p_\theta(x_{t-1} | x_t)$ is assumed as normal-distributed (with diagonal covariance matrix) with learnable parameters. Using a particular parametrization of $p_\theta(x_{t-1} | x_t)$, Ho et al. (2020) showed that the reverse process can be trained using the following objective,

$$L = \min_\theta \mathbb{E}_{x_0 \sim \mathcal{D}, \epsilon \sim \mathcal{N}(0, \mathbb{1}), t \sim \mathcal{U}(1, T)} ||\epsilon - \epsilon_\theta(\sqrt{\alpha_t} x_0 + (1 - \alpha_t)\epsilon, t)||_2^2 \,, \tag{3}$$

where $\mathcal{D}$ refers to the data distribution and $\epsilon_\theta(x_t, t)$ is parameterized using a neural network, which is equivalent to earlier score-matching techniques (Song & Ermon, 2019; Song et al., 2021). This objective can be seen as a weighted variational bound on the negative log-likelihood that down-weights the importance of terms at small $t$, i.e., at small noise levels.

Extending the unconditional diffusion process described so far, one can consider conditional variants where the backward process is conditioned on additional information, i.e. $\epsilon_\theta = \epsilon_\theta(x_t, t, c)$, where the precise nature of the conditioning information $c$ depends on the application at hand and ranges from global to local information such as spectrograms (Kong et al., 2021), see also (Pandey et al., 2022) for different ways of introducing conditional information into the diffusion process. In our case, it is given by the concatenation of input (masked according to the imputation mask) and the imputation mask itself during the backward process, i.e., $c = \text{Concat}(x_0 \odot (m_{\text{imp}} \odot m_{\text{mvi}}), (m_{\text{imp}} \odot m_{\text{mvi}})$, where $\odot$ denotes point-wise multiplication, $m_{\text{imp}}$ is the imputation mask and $m_{\text{mvi}}$ is the missing value mask. In this work, we consider two different setups, denoted as $D_0$ and $D_1$, respectively, where we apply the diffusion process to the full signal or to the regions to be imputed only. In any case, the evaluation of the loss function in Table 3 is only supposed to be on the input values for which ground truth information is available, i.e., where $m_{\text{mvi}} = 1$. For $D_0$, this can be seen as a reconstruction loss for the input values corresponding to non-zero portions of the imputation mask (where conditioning is available) and an imputation loss corresponding to input tokens at which the imputation mask vanishes, c.f. also (Du et al., 2022). For $D_1$, the reconstruction loss vanishes by construction. Finally, we also investigate an approach using a model trained in an unconditional fashion, where the conditional information is only included during inference (Lugmayr et al., 2022). In Appendix A.1, we provide a detailed comparison to CSDI(Tashiro et al., 2021), as the only other diffusion-based imputer in the literature, which crucially relates to the design decision whether to introduce a separate diffusion axis.

## 2.3 State space models

The recently introduced structured state-space model (SSM) (Gu et al., 2022a) represents a promising modeling paradigm to efficiently capture long-term dependencies in time series data. At its heart, the formalism draws on a linear state space transition equation, connecting a one-dimensional input sequence $u(t)$ to a one-dimensional output sequence $y(t)$ via a $N$-dimensional hidden state $x(t)$. Explicitly, this transition equation reads

$$x'(t) = Ax(t) + Bu(t) \text{ and } y(t) = Cx(t) + Du(t), \tag{4}$$

where $A, B, C, D$ are transition matrices. After discretization, the relation between input and output can be written as a convolution operation that can be evaluated efficiently on modern GPUs (Gu et al., 2022a). The ability to capture long-term dependencies relates to a particular initialization of $A \in \mathbb{R}^{N \times N}$ according to HiPPO theory (Gu et al., 2020; 2022b). In (Gu et al., 2022a), the authors put forward a Structured State Space sequence model (S4) by stacking several copies of the above SSM blocks with appropriate normalization layers, point-wise fully-connected layers in the style of a transformer layer, demonstrating excellent performance on various sequence classification tasks. In fact, the resulting S4 layer parametrizes a shape-preserving mapping of data with shape (batch, model dimension, length dimension) and can therefore be used as a drop-in replacement for transformer, RNN, or one-dimensional convolution layers (with appropriate padding). Building on the S4 layer, the authors presented SaShiMi, a generative model architecture for sequence generation (Goel et al., 2022) obtained by combining S4 layers in a U-Net-inspired configuration. While the model was proposed as an autoregressive model, the authors already pointed out the ability to use the (non-causal) SaShiMi as a component in state-of-the-art non-autoregressive models such as DiffWave (Kong et al., 2021).

These models as well as the proposed approaches from Section 2.4 stand in a longer tradition of approaches that rely on combinations of state space models and deep learning. These have been used for diverse tasks, such as for the modeling of high-dimensional video data, which also might allow for imputation Fraccaro et al. (2017); Klushyn et al. (2021), for causal generative temporal modeling Krishnan et al. (2015), and for forecasting Rangapuram et al. (2018); Wang et al. (2019); Li et al. (2021).

## 2.4 Proposed approaches

We propose and investigate a number of different diffusion imputer architectures. It is worth stressing that there is no prior work on direct applications of conditional diffusion models for time series imputation except

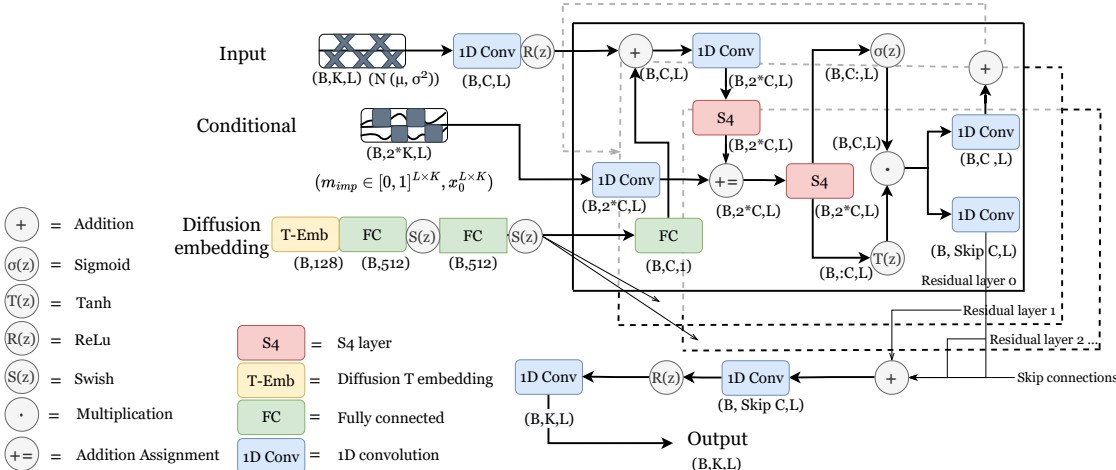

Figure 2: Proposed SSSD$^{S4}$ model architecture.

for (Tashiro et al., 2021), which is based on DiffWave, which was proposed in the context of speech synthesis. Thus, we propose a direct adaptation of the DiffWave-architecture (Kong et al., 2021) as a baseline for time series imputation, see Appendix C for details. In particular, we use the proposed setup where the diffusion process is only applied to the parts of the sequence that are supposed to imputed. Based on this model, we put forward SSSD$^{S4}$, the main architectural contribution of this work, where instead of bidirectional dilated convolutions, we used S4 layer as a diffusion layer within each of its residual blocks after adding the diffusion embedding. As a second modification, we include a second S4 layer after the addition assignment with the conditional information, which gives the model additional flexibility after joining processed inputs and the conditional information. The effectiveness of this modification is demonstrated in an ablation study in Appendix B.1. The architecture is depicted schematically in Figure 2. Second, under the name SSSD$^{SA}$ we explore an extension of the non-autoregressive of the SaShiMi architecture for time series imputation through appropriate conditioning. Third, we investigate CSDI$^{S4}$, a modification of the CSDI (Tashiro et al., 2021) architecture, where we replace the transformer layer operating in the time direction by an S4 model. In this way, we aim to assess potential improvements of an architecture that is more adapted to the domain of time series. A more detailed discussion of the model architectures including hyperparameter settings can be found in Appendix C. By construction, our imputation models always use bidirectional context (even though this could be circumvented by the use of causal S4 layers). There are situations where this use of bidirectional context could lead to data leakage, e.g. when using the imputed data as input for a forecasting task as downstream application.

At this point, we recapitulate the main components of the proposed approach: The model architecture was described in this very section, which crucially relies on the ability to use state-space model layers, see Section 2.3, as a direct replacement for convolutional layers in the original Diffwave architecture. The loss function is described in Equation (3). As described in Section 2.2, we only apply noise to the portions of the time series to be imputed with training and sampling procedures described explicitly along with pseudocode in Appendix A.4

## 3 Related Work

**Deep-learning based time series imputation**    Time series imputation is a very rich topic. A complete discussion– even of the deep learning literature alone– is clearly beyond the scope of this work, and we refer the reader to a recent review on the topic (Fang & Wang, 2020). Deep-learning-based time series imputation

methods can be broadly categorized based on the technology used: (1) RNN-based approaches such as BRITS (Cao et al., 2018), GRU-D (Che et al., 2016), NAOMI (Liu et al., 2019), and M-RNN (Yoon et al., 2019) use single or multi-directional RNNs to model the time series. However, these algorithms suffer of diverse training limitation, and as pointed out in recent works, many of them might only show sub-optimal performance across diverse missing scenarios and at different missing ratios (Cini et al., 2022) (Du et al., 2022). (2) Generative models represent the second dominant approach in the field. This includes GAN-based approaches such as $E^2$-GAN (Luo et al., 2019), and GRUI-GAN (Luo et al., 2018) or VAE-based approaches such as GP-VAE (Fortuin et al., 2020). Many of these were found to suffer from unstable training and failed to reach state-of-the-art performance (Du et al., 2022). Recently, also diffusion models such as CSDI (Tashiro et al., 2021), the closest competitor to our work, were explored with very strong results. (3) Finally, there is a collection of approaches relying in modern architectures such as graph neural networks (GRIN) (Cini et al., 2022), and permutation equivariant networks (NRTSI) (Shan et al., 2021), *self-attention* to capture temporal and feature correlations (SAITS) (Du et al., 2022), controlled differential equations networks (NeuralCDE) (Morrill et al., 2021) and ordinal differential equations networks (Latent-ODE) (Rubanova et al., 2019).

**Conditional generative modeling with diffusion models**  Diffusion models have been used for related tasks such as inpainting, in particular in the image domain (Saharia et al., 2021; Lugmayr et al., 2022). With appropriate modifications, such methods from the image domain are also directly applicable in the time series domain. Sound is a very special time series, and diffusion models such as DiffWave (Kong et al., 2021) conditioned on different global labels or Mel spectrograms showed excellent performance in different speech generation tasks. Returning to general time series, as already discussed above, the closest competitor is CSDI (Tashiro et al., 2021). CSDI and this work represent diffusion models and can be seen as DiffWave-variants. The main differences between the two approaches are (1) using SSMs instead of transformers (2) the conceptually more straightforward setup of a diffusion process in time direction only as opposed to feature and time direction, see a further discussion of it in Appendix A.1. (3) a different training objective to denoise just the segments to be imputed ($D_1$). In all of our experiments, we compare to CSDI to demonstrate our approach's superiority and that exchanging single components (as in CSDI$^{S4}$) is not sufficient to reach a qualitative improvement of the sample quality in certain BM scenarios.

**Time series forecasting**  The literature on time series forecasting is even richer than the literature on time series imputation. One large body of works includes recurrent architectures, such as LSTNet (Lai et al., 2018), and LSTMa (Bahdanau et al., 2015). Recently, also modern transformer-based architectures with encoder-decoder design such as Autoformer (Wu et al., 2021), and Informer (Zhou et al., 2021), showed excellent performance on long-sequence forecasting. The range of methods that have been applied to this task is very diverse and includes for example GP-Copula (Salinas et al., 2019), Transformer MAF (Rasul et al., 2021b), and TLAE (Nguyen & Quanz, 2021). Finally, with Time-Grad (Rasul et al., 2021a) there is another diffusion model that showed good performance on various forecasting tasks. However, due to its autoregressive nature its domain of applicability cannot be straightforwardly extended to include time series imputation.

## 4 Experiments

### 4.1 Experimental protocol

As already discussed above, we do not keep the (input) channel dimension as an explicit dimension during the diffusion process but only keep it implicitly by mapping the channel dimension to the diffusion dimension. This is inspired by the original DiffWave approach, which was designed for single-channel audio data. As we will demonstrate below, this approach leads to outstanding results in scenarios where the number of input channels remains limited to less than about 100 input channels, which covers for example typical single-patient sensor data such as electrocardiogram (ECG) or electroencephalogram (EEG) in the healthcare domain. For more input channels, the model often shows convergence issues and one has to resort to different training strategies, e.g., by splitting the input channels. As we will also demonstrate below, this approach without any further modifications already leads to competitive results on data sets with a few hundred input channels, but can certainly be improved by more elaborate procedures and is therefore not within the main

scope of this work. For this reason also the mains experimental evaluation of our work focuses primarily on data sets with less than 100 input channels, Section A.2 in the supplementary material contains a more detailed overview of the channel split approach and provide further research directions on the same matter.

Throughout this section, we always train and evaluate imputation models on identical missingness scenarios and ratios, e.g., we train on 20% RM and evaluate based on the same setting. The training on the models was performed on single NVIDIA A30 cards. We use for all of our experiments MSE as loss function, similarly, different batch sizes for each dataset implemented, see details Appendix D for details. In our experiments, we cover a very broad spectrum of popular data sets both for imputation and forecasting along with a corresponding diverse selection of baseline methods from different communities, to demonstrate the robustness of our proposed approach. There are diverse performance metrics utilized in this work (where in all cases, lower scores signify better imputation results), most of them involve comparing single imputations to the ground truth, others, incorporate imputations distribution and are therefore specific to probabilistic imputers. We refer to the supplementary material for a discussion on them. Similarly, we present small and concise tables in the main text to support our claims. We refer to the supplementary material for more results, including additional baselines, further details on data sets, and preprocessing procedures.

As a general remark, we try to quote baseline results directly from the respective original publications to avoid misleading conclusions due to sub-optimal hyperparameter choices while training baseline models ourselves. Quite naturally, this leads to a rather in-homogeneous set of baseline models available for the different datasets/tasks. To ensure better comparability, we nevertheless decided to train a single strong baseline across all tasks for both diffusion and imputation, where we explicitly mark baseline results obtained by ourselves. We chose CSDI (Tashiro et al., 2021) for this purpose, which is, on the one hand, our closest competitor as the only non-autoregressive diffusion model and, on the other hand, which represents a strong baseline both for imputation and forecasting, see the original publication. In terms of metrics, we report the commonly used metrics for the respective datasets.

## 4.2 Time series imputation

**$SSSD^{S4}$ outperforms state-of-the-art imputers on ECG data**  As first data set, we consider ECG data from the *PTB-XL data set* (Wagner et al., 2020a;b; Goldberger et al., 2000). ECG data represents an interesting benchmark case as producing coherent imputations beyond the random missing scenario requires to capture the consistent periodic structure of the signal across several beats. We preprocessed the ECG signals at a sampling rate of 100 Hz and considered $L = 250$ time steps (248 in the case of $SSSD^{SA}$). We considered three different missingness scenarios, RM, RBM, and BM. We present the investigated diffusion model variants, such as and a conditional adaptation of the original DiffWave model, $CSDI^{S4}$, $SSSD^{SA}$, and $SSSD^{S4}$, where applicable both for training objectives $D_0$ and $D_1$. As baselines we consider the deterministic LAMC (Chen et al., 2021) and CSDI (Tashiro et al., 2021) as a strong probabilistic baseline. We report averaged MAE, RMSE for 100 samples generated for each sample in the test set.

Table 1 contains the empirical results of the time series imputation experiments on the PTB-XL dataset. Across all model types and missingness scenarios, applying the diffusion process to the portions of the sample to be imputed ($D_1$) consistently yields better results than the diffusion process applied to the entire sample($D_0$). Also the unconditional training approach from Lugmayr et al. (2022) proposed in the context of a state-of-the-art diffusion-based method for image inpainting, lead to clearly inferior performance, see Appendix B.1. In the following, we will therefore restrict ourselves to the $D_1$ setting. The proposed $SSSD^{S4}$ outperforms the rest of the imputer models by a significant margin in most scenarios, in particular for BM, where we find a reduction in MAE of more than 50% compared to CSDI. Similarly, we note that DiffWave as a baseline shows very strong results, which are in some scenarios on par with the technically more advanced $SSSD^{SA}$, nevertheless, the proposed $SSSD^{S4}$ demonstrate a clear improvement for time series imputation and generation across all the settings. Also, there is a clear improvement that the S4 layer on $CSDI^{S4}$ provides to CSDI across RM and BM settings, especially for the RM scenario where $CSDI^{S4}$ outperforms the rest of methods with lower MAE. We hypothesize that the CSDI-approach is helpful in the RM setting, where consistency across features and time has to be reached, whereas $SSSD^{S4}$ and its variants show clear advantages for RBM and BM (and TF as discussed below) where modeling the time dependence is of primary importance.

Table 1: Imputation for RM, RBM and BM scenarios on the PTB-XL data set (extracts, see Table 24 for further baseline and settings comparisons, and Table 7 for missingness ratio 50%). All results (including baselines) were obtained from models trained by us.

| Model | MAE | RMSE |
|---|---|---|
| **20% RM on PTB-XL** | | |
| LAMC | 0.0678 | 0.1309 |
| CSDI | 0.0038±2e-6 | 0.0189±5e-5 |
| DiffWave | 0.0043±4e-4 | 0.0177±4e-4 |
| **CSDI$^{S4}$** | **0.0031±1e-7** | 0.0171±6e-4 |
| SSSD$^{SA}$ | 0.0045±3e-7 | 0.0181±4e-6 |
| **SSSD$^{S4}$** | 0.0034±4e-6 | **0.0119±1e-4** |
| **20% RBM on PTB-XL** | | |
| LAMC | 0.0759 | 0.1498 |
| CSDI | 0.0186±1e-5 | 0.0435±2e-4 |
| DiffWave | 0.0250±1e-3 | 0.0808±5e-3 |
| CSDI$^{S4}$ | 0.0222±2e-5 | 0.0573±1e-3 |
| SSSD$^{SA}$ | 0.0170±1e-4 | 0.0492±1e-2 |
| **SSSD$^{S4}$** | **0.0103±3e-3** | **0.0226±9e-4** |
| **20% BM on PTB-XL** | | |
| LAMC | 0.0840 | 0.1171 |
| CSDI | 0.1054±4e-5 | 0.2254±7e-5 |
| DiffWave | 0.0451±7e-4 | 0.1378±5e-3 |
| CSDI$^{S4}$ | 0.0792±2e-4 | 0.1879±1e-4 |
| SSSD$^{SA}$ | 0.0435±3e-3 | 0.1167±1e-2 |
| **SSSD$^{S4}$** | **0.0324±3e-3** | **0.0832±8e-3** |

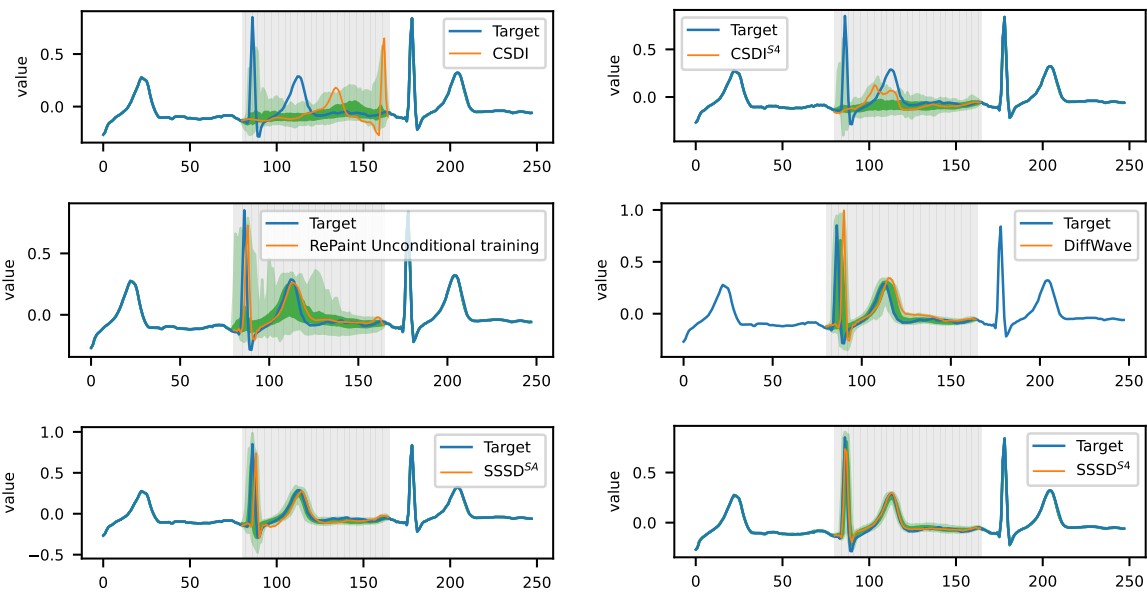

Figure 3: PTB-XL BM imputations for the V5 lead of an ECG from a healthy patient.

**Existing approaches fail to produce meaningful BM imputations** Figure 3 shows imputations on a BM scenario for a subset of models from the PTB-XL imputation task. The main purpose is to demonstrate that the achieved improvements through the proposed approaches lead to superior samples in a way that is

even apparent to the naked eye. The top figure demonstrates that the state-of-the-art imputer is unable to produce any meaningful imputations (as visible both from the shaded quantiles as well as from the exemplary imputation). As an example, the identification of a QRS complex with a duration inside the range of 0.08 sec to 0.12 sec is considered as normal signals. However, the model fails to detect the complex and rather shows a misplaced R peak. The imputation quality improves qualitatively with the proposed CSDI$^{S4}$ variant but still misses essential signal features. Only the proposed DiffWave imputer, SSSD$^{SA}$ and SSSD$^{S4}$ models capture all essential signal features. The qualitatively and quantitatively best result is achieved by SSSD$^{S4}$, which excels at the task and shows well-controlled quantile bands as expected for a normal ECG sample. We present additional experiments at a larger missingness ratio of 50% in Appendix B.2 to underline the robustness of our findings. In addition, we investigate in Appendix B.4 the robustness of the SSSD$^{S4}$ model against mismatches between training and test time in terms of missingness scenario and missingness ratio, which is of crucial importance for practical applications. The established benchmarking tasks for imputation tasks are very much focused on the RM scenario.

**SSSD$^{S4}$ shows competitive imputation performance compared to state-of-the-art approaches on other data sets and high missing ratios**  To demonstrate that the excellent performance of SSSD$^{S4}$ extends to further data sets, we collected the MuJoCo data set (Rubanova et al., 2019) from Shan et al. (2021) to test SSSD$^{S4}$ on highly sparse RM scenarios such as 70%, 80%, and 90%, for which we compare performance against the baselines RNN GRU-D (Che et al., 2016), ODE-RNN (Rubanova et al., 2019), NeuralCDE (Morrill et al., 2021), Latent-ODE (Rubanova et al., 2019), NAOMI (Liu et al., 2019), NRTSI (Shan et al., 2021), and CSDI (Tashiro et al., 2021). We report an averaged MSE for a single imputation per sample on the test set over 3 trials. All baselines results were collected from Shan et al. (2021) except for CSDI, which we trained ourselves.

Table 2: Imputation MSE results for the MuJoCo data set. Here, we use a concise error notation where the values in brackets affect the least significant digits e.g. 0.572(12) signifies $0.572 \pm 0.012$. Similarly, all MSE results are in the order of 1e-3.

| Model | 70% RM | 80% RM | 90% RM |
|---|---|---|---|
| RNN GRU-D | 11.34 | 14.21 | 19.68 |
| ODE-RNN | 9.86 | 12.09 | 16.47 |
| ODE-RNN | 9.86 | 12.09 | 16.47 |
| NeuralCDE | 8.35 | 10.71 | 13.52 |
| Latent-ODE | 3.00 | 2.95 | 3.60 |
| NAOMI | 1.46 | 2.32 | 4.42 |
| NRTSI | 0.63 | 1.22 | 4.06 |
| **CSDI** | **0.24(3)** | **0.61(10)** | 4.84(2) |
| **SSSD$^{S4}$** | 0.59(8) | 1.00(5) | **1.90(3)** |

Table 2 shows the empirical RM results on the MuJoCo data set, where SSSD$^{S4}$ outperforms all baselines except for CSDI on the 70% and 80% missingness ratio. For the scenario with highest RM ratio, 90%, SSSD$^{S4}$ outperforms all the baselines, achieving an error reduction of more than 50% with a MSE of 1.90e-3. It is worth stressing that among the three considered missingness scenarios for imputation CSDI is best-suited for RM, see Table 1 and the discussion in Appendix A.1.

**SSSD$^{S4}$ performance on high-dimensional data sets**  We also explore the potential of SSSD$^{S4}$ on data sets with more than 100 channels following the simple but potentially sub-optimal channel splitting strategy described above. We implemented the RM imputation task on the Electricity data set (Dua & Graff, 2017) from Du et al. (2022) which contains 370 features at different missingness ratios such as 10%, 30% and 50%. As baselines, we consider M-RNN (Yoon et al., 2019), GP-VAE (Fortuin et al., 2020), BRITS (Cao et al., 2018), SAITS (Du et al., 2022), a transformer variant from Du et al. (2022), and CSDI (Tashiro et al., 2021). We report an averaged MAE, RMSE, and MRE from one sample generated per test sample over a 3 trial period.

Table 3: RM Imputation results for the Electricity data set. Overall, CSDI$^{S4}$ outperformed all the RM scenarios' baselines metrics even at high levels of missing data. All baseline results were collected from Du et al. (2022).

| Model | MAE | RMSE | MRE |
|---|---|---|---|
| 10% RM on Electricity | | | |
| Median | 2.056 | 2.732 | 110% |
| M-RNN | 1.244 | 1.867 | 66.6% |
| GP-VAE | 1.094 | 1.565 | 58.6% |
| BRITS | 0.847 | 1.322 | 45.3% |
| Transformer | 0.823 | 1.301 | 44.0% |
| SAITS | 0.735 | 1.162 | 39.4% |
| CSDI | 1.510±3e-3 | 15.012±4e-2 | 81.10%±1e-3 |
| **SSSD$^{S4}$** | **0.345±1e-4** | **0.554±5e-5** | **18.4%±5e-5** |
| 30% RM on Electricity | | | |
| Median | 2.055 | 2.732 | 110% |
| M-RNN | 1.258 | 1.876 | 67.3% |
| GP-VAE | 1.057 | 1.571 | 56.6% |
| BRITS | 0.943 | 1.435 | 50.4% |
| Transformer | 0.846 | 1.321 | 45.3% |
| SAITS | 0.790 | 1.223 | 42.3% |
| CSDI | 0.921±8e-3 | 8.372±7e-2 | 49.27%±4e-3 |
| **SSSD$^{S4}$** | **0.407±5e-4** | **0.625±1e-4** | **21.8±0%** |
| 50% RM on Electricity | | | |
| Median | 2.053 | 2.728 | 109% |
| M-RNN | 1.283 | 1.902 | 68.7% |
| GP-VAE | 1.097 | 1.572 | 58.8% |
| BRITS | 1.037 | 1.538 | 55.5% |
| Transformer | 0.895 | 1.410 | 47.9% |
| SAITS | 0.876 | 1.377 | 46.9% |
| **CSDI** | **0.278±4e-3** | 2.371±3e-2 | **14.93%±1e-3** |
| **SSSD$^{S4}$** | 0.532±1e-4 | **0.821±1e-4** | 28.5%±1e-4 |

Table 3 contains the empirical results of the imputation task, where for the 10% and 30% scenarios. Overall, SSSD$^{S4}$ excelled at the imputation task, demonstrating significant error reductions against the strongest baseline SAITS and CSDI with outstanding error reductions of more than 50%, and almost 50% for each metric on the 10% and 30% scenarios respectively. For the 50% scenario, SSSD$^{S4}$ outperforms all the baselines on RMSE, while CSDI shows better performance in terms of MAE and MRE. Similarly, we tested on a 25% RM task from Cini et al. (2022) on the PEMS-Bay and METR-LA data sets (Li et al., 2018) which contains 325 and 207 features respectively. On the PEMS-Bay data set, SSSD$^{S4}$ outperformed well-established baselines such as MICE (White et al., 2011), rGAIN (Miao et al., 2021), BRITS (Cao et al., 2018), and MPGRU (Huang et al., 2019) in terms of all three metrics MAE, MSE, and MRE, while it was only superseded by the recently proposed GRIN (Cini et al., 2022). On the METR-LA data set, SSSD$^{S4}$is again outperformed by GRIN but on par with the remaining models. We refer to Table 25 in the supplementary material for a more in-depth discussion of the results.

### 4.3   Time series forecasting

**SSSD$^{S4}$ on the proposed data set**   We implemented the CSDI, CSDI$^{S4}$, SSSD$^{SA}$, and SSSD$^{S4}$ models on two data sets. For both, we report MAE, and RMSE as metrics for hundred samples generated per test sample in three trials. As first application, we reconsider the case of ECG data from the *PTB-XL data set*. As before, we work at a sampling frequency of 100 Hz, however, for this task at $L = 1000$ time steps per

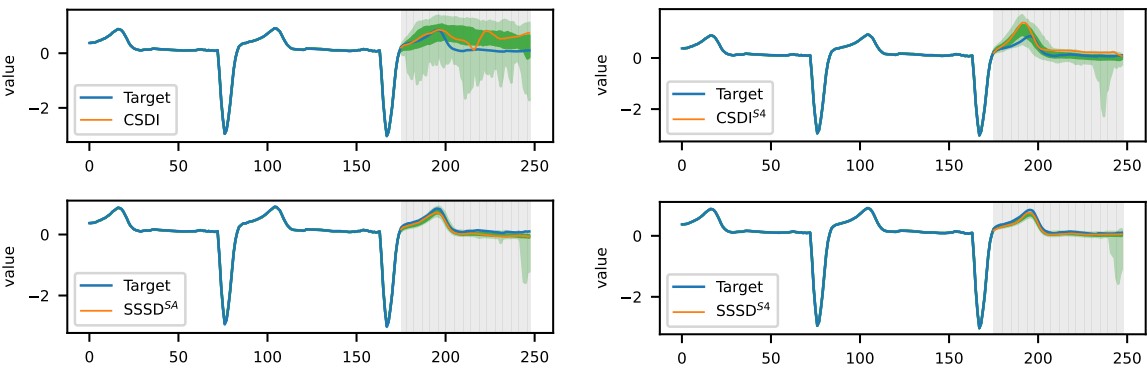

Figure 4: PTB-XL TF for the V1 lead of an ECG from a patient with a complete left bundle branch block (CLBBB).

sample, for which we condition on 800 time steps and forecast on 200. SSSD$^{SA}$ outperformed on MAE with 0.087, while SSSD$^{S4}$ achieved a slightly larger error of 0.090. SSSD$^{S4}$ outperformed on RMSE with 0.219, CSDI$^{S4}$ achieved smaller errors than CSDI on the three metrics. Again, the samples presented in Figure 4 demonstrate a clear improvement that is again even visible to the naked eye.

**SSSD$^{S4}$ shows competitive forecasting performance compared to state-of-the-art approaches on various data sets**  We test on the *Solar* data set collected from GluonTS (Alexandrov et al., 2020), a forecasting task where the conditional values and forecast horizon are 168 and 24 time steps respectively. As baselines, we consider CSDI (Tashiro et al., 2021), GP-copula (Salinas et al., 2019), Transformer MAF (TransMAF) (Rasul et al., 2021b), and TLAE (Nguyen & Quanz, 2021). All baseline results were collected from its respective original publications. We report an averaged MSE of 100 samples generated per test sample over three trials.

Table 4: Time series forecasting results for the solar data set.

| Model | MSE |
|---|---|
| GP-copula | 9.8e2 $\pm$ 5.2e1 |
| TransMAF | 9.3e2 |
| TLAE | 6.8e2 $\pm$ 7.5e1 |
| CSDI | 9.0e2 $\pm$ 6.1e1 |
| **SSSD$^{S4}$** | **2.7e2 $\pm$ 0.43e1** |

Table 4 contains the empirical results for the solar data set, which demonstrate the excellent forecasting capabilities of SSSD$^{S4}$. It achieved a MSE of 2.7e2 corresponding to a 60% error reduction compared to TLAE as strongest baseline and is also considerably stronger than CSDI, which serves as consistent baseline across all experiments.

Finally, we demonstrate SSSD$^{S4}$'s forecasting capabilities on conventional benchmark data sets for long-horizon forecasting. To this end, we collected the preprocessed ETTm1 data set from Zhou et al. (2021) and used it for forecasting at five different forecasting settings, where the forecasting length is of 24, 48, 96, 288 and 672 time steps, and the conditional values are 96, 48, 284, 288, and 384 time steps, respectively. We compare to LSTnet (Lai et al., 2018), LSTMa (Bahdanau et al., 2015), Reformer (Kitaev et al., 2020), LogTrans (Li et al., 2019), Informer (Zhou et al., 2021), one Informer baseline called Informer(†), Autoformer (Wu et al., 2021), and CSDI (Tashiro et al., 2021). We report an averaged MAE and MSE for a single sample generated for each test sample over 2 trials. All baseline results were collected from Zhou et al. (2021); Wu et al. (2021) except for CSDI, which we trained ourselves.

Table 5: Time series forecasting results on the ETTm1 data set.

| Model | 24 | 48 | 96 | 288 | 672 |
|---|---|---|---|---|---|
| TF on ETTm1 (MAE) | | | | | |
| LSTNet | 1.170 | 1.215 | 1.542 | 2.076 | 2.941 |
| LSTMa | 0.629 | 0.939 | 0.913 | 1.124 | 1.555 |
| Reformer | 0.607 | 0.777 | 0.945 | 1.094 | 1.232 |
| LogTrans | 0.412 | 0.583 | 0.792 | 1.320 | 1.461 |
| Informer† | 0.371 | 0.470 | 0.612 | 0.879 | 1.103 |
| Informer | 0.369 | 0.503 | 0.614 | 0.786 | 0.926 |
| CSDI | 0.370(3) | 0.546(2) | 0.756(11) | 0.530(4) | 0.891(37) |
| **Autoformer** | 0.403 | **0.453** | **0.463** | **0.528** | **0.542** |
| **SSSD$^{\text{S4}}$** | **0.361(6)** | 0.479(8) | 0.547(12) | 0.648(10) | 0.783(66) |
| TF on ETTm1 (MSE) | | | | | |
| LSTNet | 1.968 | 1.999 | 2.762 | 1.257 | 1.917 |
| LSTMa | 0.621 | 1.392 | 1.339 | 1.740 | 2.736 |
| Reformer | 0.724 | 1.098 | 1.433 | 1.820 | 2.187 |
| LogTrans | 0.419 | 0.507 | 0.768 | 1.462 | 1.669 |
| **Informer†** | **0.306** | 0.465 | 0.681 | 1.162 | 1.231 |
| Informer | 0.323 | 0.494 | 0.678 | 1.056 | 1.192 |
| **CSDI** | 0.354(15) | 0.750(4) | 1.468(47) | **0.608(35)** | 0.946(51) |
| **Autoformer** | 0.383 | **0.454** | **0.481** | 0.634 | **0.606** |
| SSSD$^{\text{S4}}$ | 0.351(9) | 0.612(2) | 0.538(13) | 0.797(5) | 0.804(45) |

Table 5 contains forecasting results on the ETTm1 dataset. The experimental results confirm again SSSD$^{\text{S4}}$'s robust forecasting capabilities, specifically for long-horizon forecasting, where also the conditional time steps increases. For the first setting, SSSD$^{\text{S4}}$ outperformed the rest of the baselines on MAE. For the second setting, on shorter forecast conditional and target lengths SSSD$^{\text{S4}}$ scores are comparable with Autoformer and Informer, while on the remaining three settings, SSSD$^{\text{S4}}$ outperformed all baselines with in parts significant error reductions but does not reach the performance of Autoformer. SSSD$^{\text{S4}}$ performs consistently better than CSDI except for a forecasting length of 288, where CSDI even outperforms Autoformer. It is worth stressing that Autoformer represents a very strong baseline, which was tailored to time series forecasting including a decomposition into seasonal and long-term trends, which is perfectly adapted to the forecasting task on the ETTm1. It is an interesting question for future research if SSSD$^{\text{S4}}$ can also profit from a stronger inductive bias and/or if there are forecasting scenarios, which do not follow the decomposition into seasonal and long-term trends so perfectly, where such a kinds of inductive bias can even be harmful. To summarize, we see it as a very encouraging sign that a versatile method such as SSSD$^{\text{S4}}$, which is capable of performing imputation as well as forecasting, is clearly competitive with most of the forecasting baselines.

## 5 Conclusion

In this work, we proposed the combination of structured state space models as emerging model paradigm for sequential data with long-term dependencies and diffusion models as the current state-of-the-art approach for generative modeling. The proposed SSSD$^{\text{S4}}$outperforms existing state-of-the-art imputers on various data sets under different missingness scenarios, with a particularly strong performance in blackout missing and forecasting scenarios, provided the number of input channels does not grow too large. In particular, we present examples where the qualitative improvement in imputation quality is even apparent to the naked eye. We see the proposed technology as a very promising technology for generative models in the time series domain, which opens the possibility to build generative models conditioned on various kinds of information from global labels to local information such as semantic segmentation masks, which in turn enables a broad range of further downstream applications. The source code underlying our experiments is available under https://github.com/AI4HealthUOL/SSSD.

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

# A  Technical and implementation details

## A.1  Time and feature diffusion

In this section we compare to the different design choices between the proposed approach and CSDI (Tashiro et al., 2021), as the only other diffusion-based imputer in the literature. As in (Tashiro et al., 2021), we base our parametrization of $\epsilon_\theta(x_t, t, c)$ on the DiffWave architecture (Kong et al., 2021) as a versatile diffusion model architecture proposed in the context of audio generation.

CSDI relies on introducing a separate diffusion dimension into the original three-dimensional input representation, i.e. working on an extended four-dimensional internal representation of shape (batch dimension, diffusion dimension, input channel dimension, time dimension). This necessitates processing time and feature dimensions alternatingly since many modern architectures for sequential data, such as transformers or structured state space models, are only able to process sequential, i.e. three-dimensional, input batches.

We take the conceptually simpler path of mapping the input channels into the diffusion dimension and performing only diffusion along the time dimension, i.e. processing batches of shape (batch dimension, diffusion dimension, time dimension), which implies that no separate diffusion process along time and feature axis are required as a single diffusion along the time direction is sufficient. This choice also corresponds to the common approach when applying diffusion models for example to image or sound data.

We see this design choice as central explanation for the superior performance of SSSD$^{S4}$ compared to CSDI (as well as CSDI$^{S4}$, which should perform on par with SSSD$^{S4}$ if the improved handling of the time diffusion was the major issue) in the regime of a small number of input channels. We hypothesize that for a large number of input channels it becomes increasingly difficult for SSSD$^{S4}$ to reconstruct the original input channels from the internal diffusion channels. We see the central strength of the proposed model in its ability to capture long-term dependencies along the time direction, as required for example to perform imputations in the BM scenario. The approach taken by CSDI of disentangling feature and diffusion axes, seems to provide advantages in situations with many input channels and/or situations, such as RM imputation, where the temporal consistency is less important than the ability of inferring relations between different channels at a given time step.

## A.2  Channel splitting

As described in the main text, we found convergence issues during training in our proposed models which are direct variants from DiffWave such as SSSD$^{SA}$, and SSSD$^{S4}$ using more than about 100 input channels, which we attribute to the handling of the diffusion axis, see the discussion in Appendix A.1. The simplest way to generate predictions also for cases with more input channels is by splitting up original batches of shape (batch dimension, input channel dimension, time dimension) along the channel dimension into several batches of shape (batch dimension, reduced input channel dimension, time dimension), where we just used consecutive channels in the original channel order provided by the data set. This approach can be implemented straightforwardly but comes with the obvious disadvantage that identical channels in these input batches with reduced channel dimension can correspond to different channels in the original input, which might lead to suboptimal prediction performance.

Apart from addressing this issue by keeping a separate channel axis, see Appendix A.1, we see the extension of the approach to include more elaborate channel splitting procedures as an interesting direction for future research. First, batches with reduced channels could be defined based on channel correlations instead of on the original channel order based on the assumption that the model should be able to leverage information most effectively from correlated channels. Second, the information which of the different reduced channel batches was passed could be incorporated as additional information in order to learn channel-specific representations across batches that originate from different channels in the original input.

### A.3 Implementation details on missingness scenarios

As described in Section 2, our research focuses on three different masking strategies, RM, RBM, BM, as well as TF, which can be seen as a special case of BM. For RM, we randomly sample a number of missingness ratio × total time steps as imputation target, separately for each channel. For RBM and BM we start by partitioning the sequence into segments of length missingness ratio × total time steps, where a potential remainder to reach the full sequence length is assigned to a final segment. For RBM, we randomly sample one of these segments per channel as imputation target. For BM we only sample a single segment, which is used as imputation target across all channels.

### A.4 Training and Sampling Code

---
**Algorithm 1** Training step

---
**Require:** Diffusion hyperparameters $\{T, \beta_0, \beta_1\}$, Net (model), $m_{\text{imp}}$ (imputation mask), $m_{\text{mvi}}$ (missing value mask), D1 (use mode D1), $x_0$ (ground truth signal)

$\quad \beta = \text{LinSpace}(\text{start} = \beta_0, \text{end} = \beta_1, \text{steps} = T)$ $\qquad\qquad$ ▷ Compute diffusion process hyperparameters

$\quad \alpha = 1 - \beta$

$\quad \bar{\alpha}_t = \alpha$

$\quad \beta_t = \beta$

$\quad$ **for** t in range(1, T) **do**

$\quad\quad \bar{\alpha}_t = \prod_{s=1}^{t} \alpha_s$

$\quad\quad \tilde{\beta}_t = \beta_t \cdot (1 - \bar{\alpha}_{t-1})/(1 - \bar{\alpha}_t)$

$\quad\quad \sigma_t^2 = \tilde{\beta}_t$

$\quad$ **end for**

$\quad x_{std} = \mathcal{N}(0, \mathbb{1})^{C^{\text{size}}}$ $\qquad\qquad\qquad\qquad\qquad\qquad\qquad\qquad\qquad$ ▷ Training

$\quad DS = \text{RandInt}(T)^{C^{\text{size}}}$

$\quad M = m_{\text{imp}} \odot m_{\text{mvi}}$

$\quad C = \text{Concat}(x_0 \odot M, M)$

$\quad$ **if** D1 **then**

$\quad\quad x_{std} = x_0 \odot M + x_{std} \odot (1 - M)$

$\quad$ **end if**

$\quad \bar{x} = \sqrt{\bar{\alpha}_t[DS]} \times x_0 + \sqrt{(1 - \bar{\alpha}_t[DS])} \times x_{std}$

$\quad y = \text{Net}(\bar{x}, C, DS)$

$\quad$ **if** D1 **then**

$\quad\quad \text{loss} = (y \odot M - x_0 \odot M)^2$

$\quad$ **else**

$\quad\quad \text{loss} = (y - x_0)^2$

$\quad$ **end if**

$\quad \text{update\_parameters(loss)}$

---

Algorithm 1 contains pseudocode for the training step in our experiments. It starts with the computation of the diffusion process hyperparameters $\beta$, $\alpha$, $\bar{\alpha}_t$, and $\sigma_t^2$ given the number of diffusion steps $T$ and the beta schedule start $\beta_0$ and beta schedule end $\beta_1$ values between which we interpolate linearly. For the training step, firstly, we instiate random Gaussian noise of the same shape as the input batch $\mathcal{N}(0, \mathbb{1})^{C^{\text{size}}}$, and compute diffusion steps $DS$ as random integers of the size of the conditional input batch. Secondly, we create the combined mask $M$ obtained via point-wise multiplication of imputation mask $m_{\text{imp}}$ and missing value mask $m_{\text{mvi}}$. Thirdly, the conditional information $C$ is obtained by concatenating the input $x_0$ (masked according to the imputation mask) and the imputation mask itself. Then, if our diffusion setting is $D_1$, the conditional information gets indexed into the noisy batch given $M$. Then, we update $x_0$ as $\bar{x}$ from $q(\{x\|x_0\})$. Afterwards, the diffusion model outputs a generated time series given $\bar{x}$, $C$, and $DS$. Finally, we use mean squared error as loss function and update the network parameters using backpropagation.

Algorithm 2 shows pseudocode for the sampling procedure applied during our experiments. First, we again compute the diffusion process hyperparameters $\beta$, $\alpha$, $\bar{\alpha}_t$, and $\sigma_t^2$ given the number of diffusion steps $T$ and the

---

**Algorithm 2** Sampling algorithm

---

**Require:** Diffusion hyperparameters $\{T, \beta_0, \beta_1\}$, Net (model), C (conditioning; observed signal), M (combined
    mask $M = m_{\mathrm{imp}} \odot m_{\mathrm{mvi}}$), D1 (use mode D1)
    $\beta = \mathrm{LinSpace}(\mathrm{start} = \beta_0, \mathrm{end} = \beta_1, \mathrm{steps} = T)$             ▷ Compute diffusion process hyperparameters
    $\alpha = 1 - \beta$
    $\bar{\alpha}_t = \alpha$
    $\beta_t = \beta$
    **for** t in range(1, T) **do**
        $\bar{\alpha}_t = \prod_{s=1}^{t} \alpha_s$
        $\tilde{\beta}_t = \beta_t \cdot (1 - \bar{\alpha}_{t-1})/(1 - \bar{\alpha}_t)$
        $\sigma_t^2 = \tilde{\beta}_t$
    **end for**
    $C = \mathrm{Concat}(C, M)$                                                 ▷ Sampling
    $x_{std} = \mathcal{N}(0, \mathbb{1})^{C^{\mathrm{size}}}$
    **for** t in range(T-1, -1, -1) **do**
        **if** D1 **then**
            $x = x_{std} \odot (1 - M) + C \odot M$
        **end if**
        $DS = t \times \mathrm{ones}([C^{\mathrm{size}}[0], 1])$
        $\epsilon_\theta = \mathrm{Net}(x, C, DS)$
        $x = (x_0 - (1 - \alpha[t])/\sqrt{(1 - \bar{\alpha}_t[t])} \times \epsilon_\theta)/\sqrt{\alpha[t]}$
        **if** $t > 0$ **then**
            $x = x + \sigma_t^2[t] \times \mathcal{N}(0, \mathbb{1})^{C^{\mathrm{size}}}$
        **end if**
    **end for**

---

beta schedule start $\beta_0$ and beta schedule end $\beta_1$ values between which we interpolate linearly. For sampling, we start from random Gaussian noise of the same shape as batch $\mathcal{N}(0, \mathbb{1})^{C^{\mathrm{size}}}$, then, it gets updated, first, by the masking strategy in case the diffusion is applied only to the segments to be imputed, second, by the model generation which also takes as input the observed series to condition on $C$, the mask $M$, and diffusion steps $DS$, and lastly updated from $\epsilon_\theta$ to x ($x_{t-1}$ to $\mu_\theta(x_t)$) given the diffusion hyperparameters, where if the diffusion step is not the last, we add a variance term to $x_{t-1}$.

## B  Ablation studies

### B.1  Model architecture

Table 6 contains the results obtained within an ablation study done aiming to find the best configuration for an S4 layer in our proposed model. We present three different variants with capital letter notations. (A) represents our proposed DiffWave imputer with dilated convolutional layers. (B) represents a DiffWave imputer with a S4 layer in replacement of the dilated convolutional layer. (C) represents setting B and an extra S4 layer after the conditional data addition (SSSD$^{\mathrm{S4}}$). Lastly, setting (D) represents the testing of our proposed SSSD$^{\mathrm{S4}}$, however, with an unconditional training, following the procedure proposed in (Lugmayr et al., 2022) for image inpainting.

The experiment was carried out in the three investigated missing scenarios RM, RBM, and BM at two different missingness ratios, 20% and 50%BM on the PTB-XL data set. Metrics were obtained at 150,000 training iterations. We report an averaged MAE, and RMSE, over three trials for 10 samples generated for each sample of the test set. The results in Table 6 shows the metrics obtained for the four different model variants considered in our ablation study. The replacement of the dilated convolutions by S4 layers leads to a significant error reduction (setting B vs. setting A) for most of the scenarios. The introduction of a second S4 layer again leads to a smaller but still consistent improvement over the setup with a single S4 layer (setting C vs. setting B) for all of the scenarios. The unconditional training procedure proposed in (Lugmayr

Table 6: Results of the ablation study on model architecture.

| Model | MAE | RMSE | MAE | RMSE |
|---|---|---|---|---|
| | 20% RM on PTB-XL | | 50% RM on PTB-XL | |
| A: Diffwave | 4.30e-3±4e-4 | 1.77e-2±4e-4 | 9.10e-3±5e-5 | 4.25e-2±4e-4 |
| B: SSSD$^{S4}$(single S4 layer) | 4.75e-3±1e-4 | 2.01e-2±1e-4 | 8.15e-3±4e-5 | 3.64e-2±5e-4 |
| C: **SSSD$^{S4}$** | **3.40e-3±4e-6** | **1.19e-2±1e-4** | **8.00e-3±8e-5** | **3.53e-2±6e-4** |
| D: RePaint (uncond. training) | 1.34e-2±9e-5 | 3.76e-2±1e-3 | 2.31e-2±8e-5 | 6.13e-2±1e-3 |
| | 20% RBM on PTB-XL | | 50% RBM on PTB-XL | |
| A: Diffwave | 2.50e-2±1e-3 | 8.08e-2±5e-3 | 4.84e-2±3e-3 | 1.40-e1±4e-3 |
| B: SSSD$^{S4}$(single S4 layer) | 2.42e-2±5e-4 | 1.25e-1±6e-3 | 4.49e-2±2e-3 | 1.36e-1±5e-3 |
| C: **SSSD$^{S4}$** | **1.03e-2±3e-3** | **2.26e-2±9e-4** | **3.73e-2±4e-4** | **1.31e-1±3e-3** |
| D: RePaint (uncond. training) | 6.40e-2±7e-4 | 1.57e-1±1e-3 | 1.26e-1±5e-3 | 2.51e-1±1e-2 |
| | 20% BM on PTB-XL | | 50% BM on PTB-XL | |
| A: Diffwave | 4.35e-2±3e-3 | 1.16e-1±1e-2 | 1.23e-1±6e4 | 2.76e-1±3e-4 |
| B: SSSD$^{S4}$(single S4 layer) | 3.67e-2±2e-3 | 9.29e-2±2e-2 | 1.46e-1±1e-4 | 2.81e-1±4e-3 |
| C: **SSSD$^{S4}$** | **3.24e-2±3e-3** | **8.32e-2±8e-3** | **1.16e-1±3e-3** | **2.66e-1±3e-3** |
| D: RePaint (uncond. training) | 1.23e-1±4e-3 | 2.13e-1±8e-3 | 1.67e-1±1e-3 | 3.10e-1±2e-4 |

et al., 2022) is clearly inferior to the proposed conditional training (setting D vs. setting C), see also Figure 5 for a qualitative impression of the generated samples. This leads us to propose SSSD$^{S4}$ (setting C) as default variant for further experiments.

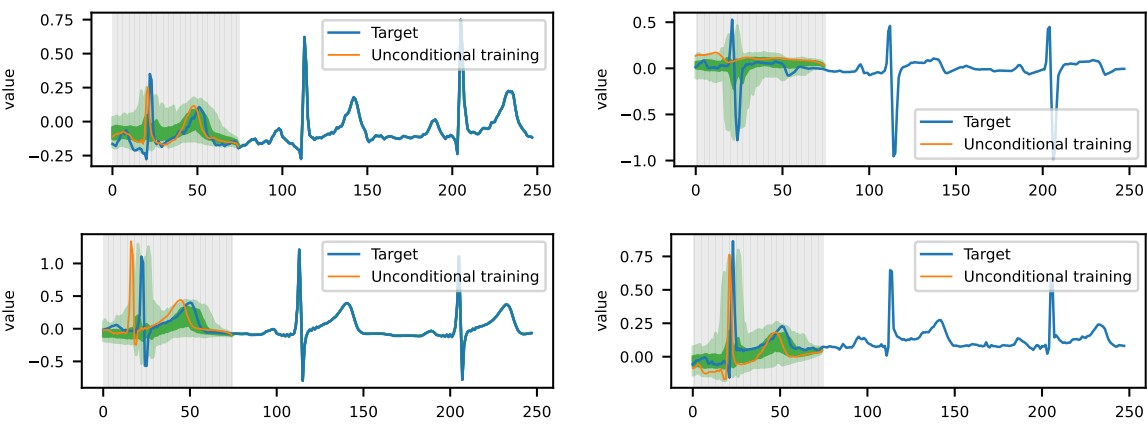

Figure 5: PTB-XL BM imputations for the leads I, V1, V4, and aVF of an ECG from a patient with sinus arrhythmia obtained from unconditional training (setting D).

## B.2 Additional missingness ratio on PTB-XL

Table 7 we present additional empirical results on the PTB-XL dataset at 50% of missigness across all investigated scenarios for a selected set of models (CSDI, DiffWave, and SSSD$^{S4}$), which complement the corresponding results in Table 1 for a missingness ratio of 20%. We observe that DiffWave (with dilated convolutional layers) outperforms CSDI on the RBM and BM missing scenarios, however, not in the RM scenario, which is best suited for the approach taken by CSDI, see the discussion in Appendix A.1. SSSD$^{S4}$ outperforms both CSDI and DiffWave for all three scenarios and both metrics in line with its strong performance demonstrated for smaller missingness ratios.

Table 7: Results for selected models at a missingness ratio of 50% on PTB-XL.

| Model | MAE | RMSE |
|---|---|---|
| 50% RM on PTB-XL | | |
| CSDI | 9.00e-3±1.00e-4 | 7.20e-2±2.30e-3 |
| DiffWave | 9.10e-3±5.77e-5 | 4.25e-2±4.58e-4 |
| **SSSD$^{S4}$** | **8.00e-3±8.81e-5** | **3.53e-2±6.82e-4** |
| 50% RBM on PTB-XL | | |
| CSDI | 6.25e-2±1.60e-3 | 1.54e-1±4.50e-3 |
| DiffWave | 4.84e-2±3.11e-3 | 1.40-e1±4.48e-3 |
| **SSSD$^{S4}$** | **3.73e-2±4.91e-4** | **1.31e-1±3.45e-3** |
| 50% BM on PTB-XL | | |
| CSDI | 1.49e-1±2.10e-3 | 2.95e-1±5.60e-3 |
| DiffWave | 1.23e-1±6.80e-4 | 2.76e-1±3.28e-4 |
| **SSSD$^{S4}$** | **1.16e-1±3.05e-3** | **2.66e-1±3.23e-3** |

## B.3 Dependence on number of diffusion steps

Table 8: Imputation for BM scenario on the PTB-XL data set at 50% BM with 50 and 200 diffusion steps.

| Model | Diffusion steps | Training time | Sampling time | MAE \| RMSE |
|---|---|---|---|---|
| SSSD$^{S4}$ | 50 | 31 hours | 627 seconds | 1.93e-1±7.00e-3 \| 0.350±1.23e-2 |
| SSSD$^{S4}$ | 200 | 36 hours | 3,740 seconds | 1.16e-1±3.05e-3 \| 0.267±3.23e-3 |

Table 8 demonstrates the impact of reducing the number of diffusion steps from the default value 200 used in this work to 50. It leads to a slightly faster training time (86% of the original training time) and to considerably faster sampling times (16% of the original sampling time), however, comes at the cost of considerably worse target metrics (166% and 131% of the original metrics). It is worth mentioning that alternative methods, such as (Meng et al., 2022), have been proposed to reduce the sampling time, which do not come with such disadvantages.

## B.4 Robustness analysis

In this ablation study we study the robustness of our models with respect to a mismatch of either missingness ratio or missingness scenario between train and test. While the latter is rather relevant for identifying optimal training scenarios, the former is also very relevant for the practical applicability of imputation models as for real-world scenarios it is not realistic to assume a constant missingness ratio.

Table 9: Imputation for BM scenario on the PTB-XL data set for the benchmarking of different missingness ratios such as 20%, and 50%.

| Setting | Test 20% | Test 50% |
|---|---|---|
| | MAE \| RMSE | |
| Train 20% BM | 3.24e-2±3.00e-3 \| 8.32e-2±8.00e-3 | 1.29e-1±1.60e-3 \| 2.71e-1±7.89e-3 |
| Train 50% BM | 8.99e-2±1.04e-3 \| 2.26e-1±4.40e-3 | 1.16e-1±3.05e-3 \| 2.66e-1±3.23e-3 |

We carry out the experiments on the effect of a mismatch in missingness ration on PTB-XL in the BM scenario for missingness ratios of 20% and 50%. The results are compiled in Table 9. They show that the model trained on smaller missingness ratio but evaluated on a larger missingness ratio takes a penalty of 11% (MAE) in comparison to model that was trained and evaluated on the high missingness ratio. In the opposite case where trained on a larger missingness ratio but evaluated at a smaller one reveals a larger

performance malus with metrics that are almost three times larger compared to the performance of a model trained at the matching small missingness ration. This limited experiment seems to indicate that models are significantly more robust against testing at higher missingness ratios compared to the missingness ratio seen at training time. However, it is also worth stressing that the mismatch between the missingness ratios is already quite large in this experiment.

Table 10: Imputation at 50% missingness ratios on the PTB-XL data set for the benchmarking of different scenarios such as RM, RBM, and BM.

| MAE | Test RM | Test RBM | Test BM |
|---|---|---|---|
| | 50% MAE | | |
| Train RM | **8.00e-3±8.81e-5** | 8.11e-2±2.49e-3 | 1.60e-1±7.59e-3 |
| Train RBM | 1.40e-2±4.99e-5 | **3.73e-2±4.91e-4** | **8.79e-2±2.25e-3** |
| Train BM | 1.49e-2±4.99e-5 | 8.41e-2±4.40e-3 | 1.16e-1±3.05e-3 |
| | 50% RMSE | | |
| Train RM | **3.53e-2±6.82e-4** | 1.78e-1±7.50e-4 | 2.69e-1±1.84e-2 |
| Train RBM | 4.94e-2±1.30e-3 | **1.31e-1±3.45e-3** | **2.17e-1±6.35e-3** |
| Train BM | 5.21e-2±1.00e-3 | 2.16e-1±3.50e-4 | 2.66e-1±3.23e-3 |

The second experiment addresses the robustness with respect to mismatches in missingness scenarios between train and test time. This seems like a less realistic scenario for practical applications where the missingness scenario is unlikely to change but touches on the important question if training on identical missingness characteristics expected during test time is really the optimal approach. In Table 10 we carry out such an experiment again on the PTB-XL dataset comparing RM, RBM and BM as missingness scenarios, for simplicity at a fixed missingness ration of 50% in all cases. For testing on RM and RBM, the respective matching training conditions clearly lead to the best results. Interestingly, for testing on BM, training on RBM leads to considerable reductions (25% in terms of MAE and 19% in terms of RMSE) compared to training on BM. This represents an interesting finding worth pursuing in future research.

## C   Training hyperparameters and architectures

Table 11: LAMC hyperparameters.

| Hyperparameter | Value |
|---|---|
| Rhos | 0.001, 0.01, 0.1, 1, 2, 3, 5, 10, 20. |
| Lambdas | 0.001, 0.01, 0.1, 1, 2, 3, 5, 6, 8, 10. |
| Epsilons | 1e-4, 1e-3, 1e-2, 1, 2, 5, 8, 10. |
| Ranks | 2, 5, 10, 20, 50, 80, 100. |
| Iterations | 100 |

**LAMC**   For the LAMC algorithm (Chen et al., 2021), we defined a range of hyperparameters to be chosen by an exhaustive grid search given the best test metrics. Table 11 contains the LAMC hyperparameter implemented through all the experiments. We decided to implement this algorithm as a baseline as it represents a qualitatively different methodology for time series imputation compared to our probabilistic diffusion models, which relies on matrix completion approach.

**CSDI and CSDI$^{S4}$**   We use the official implementation of CSDI (Tashiro et al., 2021) as a baseline with the authors' default hyperparameters settings. The only modification for CSDI$^{S4}$ was to replace the temporal transformer layer by a S4 layer. Table 12 contains all the hyperparameters that for CSDI and CSDI$^{S4}$ training and architecture. For the CSDI$^{S4}$ setting, the S4 model implemented in all experiments is a bidirectional layer with layer normalization, no dropout, and $N = 64$ as internal state dimensionality.

Table 12: CSDI hyperparameters.

| Hyperparameter | Value |
|---|---|
| Residual layers | 4 |
| Residual channels | 64 |
| Diffusion embedding dim. | 128 |
| Schedule | Quadratic |
| Diffusion steps $T$ | 50 |
| $B_0$ | 0.0001 |
| $B_1$ | 0.5 |
| Feature embedding dim. | 128 |
| Time embedding dim. | 16 |
| Self-attention layers time dim. | 1 |
| Self-attention heads time dim. | 8 |
| Self-attention layers feature dim. | 1 |
| Self-attention heads feature dim. | 8 |
| Optimizer | Adam |
| Loss function | MSE |
| Learning rate | $1 \times 10^{-3}$ |
| Weight decay | $1 \times 10^{-6}$ |

Table 13: DiffWave hyperparameters.

| Hyperparameter | Value |
|---|---|
| Residual layers | 36 |
| Residual channels | 256 |
| Skip channels | 256 |
| Diffusion embedding dim. 1 | 128 |
| Diffusion embedding dim. 2 | 512 |
| Diffusion embedding dim. 3 | 512 |
| Schedule | Linear |
| Diffusion steps $T$ | 200 |
| $B_0$ | 0.0001 |
| $B_1$ | 0.02 |
| Optimizer | Adam |
| Loss function | MSE |
| Learning rate | $2 \times 10^{-4}$ |

**DiffWave**  Table 13 contains all the hyperparameters for training and architecture of the DiffWave imputer. As previously discussed, DiffWave was released by its authors (Kong et al., 2021) as diffusion model for speech synthesis. Here, we are referring to DiffWave model as a custom implementation in the context of time series imputation/forecasting. Some of the architectural differences are a large number of residual layers and channels, but most importantly our proposed training procedure which involves a different use of conditional information, such as the concatenation conditioning information and binary mask.

**SSSD$^{\text{SA}}$**  Table 14 contains the SSSD$^{\text{SA}}$ hyperparameters for all our experiments. SSSD$^{\text{SA}}$ is a variant of the SaShiMi (Goel et al., 2022) model, a 128-dimensional U-Net model with six residual layers, consisting of an S4 and feed-forward layers in a block manner at each pooling level, where the pooling factor in decreasing the sequence length is 2 and 2 and its respective feature expansion of 2. There is a single three-layer diffusion embedding with dimensions 128, 512, and 512, respectively for all the residual layers. As for S4, we use a bidirectional layer with layer normalization, no dropout, and an internal state dimension of $N = 64$, but with a gated linear unit in each layer.

Table 14: SSSD$^{\text{SA}}$ hyperparameters.

| Hyperparameter | Value |
|---|---|
| Residual layers | 6 |
| Pooling factor | [2,2] |
| Feature expansion | 2 |
| Diffusion embedding dim. 1 | 128 |
| Diffusion embedding dim. 2 | 512 |
| Diffusion embedding dim. 3 | 512 |
| Schedule | Linear |
| Diffusion steps $T$ | 200 |
| $B_0$ | 0.0001 |
| $B_1$ | 0.02 |
| Optimizer | Adam |
| Loss function | MSE |
| Learning rate | $2 \times 10^{-4}$ |

Table 15: SSSD$^{\text{S4}}$ hyperparameters.

| Hyperparameter | Value |
|---|---|
| Residual layers | 36 |
| Residual channels | 256 |
| Skip channels | 256 |
| Diffusion embedding dim. 1 | 128 |
| Diffusion embedding dim. 2 | 512 |
| Diffusion embedding dim. 3 | 512 |
| Schedule | Linear |
| Diffusion steps $T$ | 200 |
| $B_0$ | 0.0001 |
| $B_1$ | 0.02 |
| Optimizer | Adam |
| Loss function | MSE |
| Learning rate | $2 \times 10^{-4}$ |

**SSSD$^{\text{S4}}$** We present in Table 15 the proposed SSSD$^{\text{S4}}$ hyperparameters and training settings used throughout all our experiments. The SSSD$^{\text{S4}}$ a model builds on DiffWave (Kong et al., 2021) and consists of 36 stacked residual layers with 256 residual and skip channels. As for SSSD$^{\text{SA}}$, SSSD$^{\text{S4}}$ uses a three-layer diffusion embedding of 128, 256, and 256 hidden units dimensions with a swish activation function after the second and third layer. After the addition of diffusion embedding, we implemented a convolutional layer to double the channel dimension of the input before computing the first S4 diffusion. Similarly, after a similar expansion of the conditional information and its addition to the input, there is the application of a second S4 layer. Then the output is passed through a gated tanh activation function as non-linearity, from which we then project back from residual channels to the channel dimensionality with a convolutional layer. We used 200 time steps on a linear schedule for diffusion configuration from a beta of 0.0001 to 0.02. We utilized Adam as an optimizer with a learning rate of $2 \times 10^{-4}$. For the S4 model, similar to the other approaches, we used a bidirectional layer with layer normalization, no dropout, and an internal state dimension of $N = 64$.

**S4** In Table 16 we present the hyperparameter settings for the state space S4 model (Gu et al., 2022a) used in this work. Overall, we utilize a single S4 layer with a bidirectional setting for time dependencies learning of series in both directions. Similarly, we applied a layer normalization and utilize a internal state of dimensionality $N = 64$ as used in prior work (Gu et al., 2022a).

Table 16: S4 hyperparameters.

| Hyperparameter | Value |
|---|---|
| Layers | 1 |
| State $N$ dimensions | 64 |
| Bidirectional | Yes |
| Layer normalization | Yes |
| Drop-out | 0.0 |
| Maximum length | as required |

**Hyperparameter tuning** In this paragraph, we briefly describe the strategy that led to the hyperparameter selections discussed in the previous paragraphs: As a first remark, we focus on SSSD$^{S4}$ in this section. These settings were applied to SSSD$^{SA}$ without further experimentation. We only adapted the number of residual layers to obtain a model with comparable computational complexity as SSSD$^{S4}$. For CSDI$^{S4}$, we left the hyperparameters as close to the original default hyperparameters as possible. As a general remark, we point out that the hyperparameter optimization was carried out based on validation set scores on the PTB-XL data set at 248 time steps. We did not adjust the hyperparameters for other data sets, which can be seen as a hint for the robustness of the proposed method. Below, we briefly comment on specific aspects of the hyperparameter selection.

*Diffusion hyperparameters*: We implemented a linear schedule with a minimum noise level $B_0$ of 0.001 and a maximum noise level of $B_1$ as 0.5, based on trial and error. We found that the fewer diffusion steps, the faster the network converges during training, however, at the cost of less accurate results. In this respect, using 200 diffusion steps represented a reasonable compromise. Supporting results can be found in Appendix B.3.

*Three-layer diffusion embedding*: From the beginning of our experiments, we implemented a three-layer embedding with 128, 512, and 512 hidden units, respectively. While reducing the residual layers in some experiments, we also experimented with reducing those dimensionalities to 16, 32, 32, or 64, 64, 256. However, we found that the former choice led to better results.

*Residual channels and skip channels*: we introduced a large number of channels (256) to avoid the degradation problem.

*Residual layers*: We set the number of residual layers to 36 after the experimental phase of our ablation study. In our previous experiments with a single S4 layer, we worked with 48 layers but reduced it to 36 to increase the speed of convergence during training and to reduce the computational complexity of the overall model, which in variant C requires two S4 layers.

## D  Data set descriptions

Table 17: PTB-XL data set details.

| Description / Setting | PTB-XL 248 | PTB-XL 1000 |
|---|---|---|
| Train size | 69,764 | 17,441 |
| Validation size | 8,772 | 2,193 |
| Test size | 8,812 | 2,203 |
| Training batch | 32 | 4 |
| Sample length | 248 | 1000 |
| Sample features | 12 | 12 |
| Conditional values | 198 | 800 |
| Target values | 52 | 200 |

**PTB-XL data set** Table 17 contains the PTB-XL data set details (Wagner et al., 2020a;b; Goldberger et al., 2000). The PTB-XL ECG data set consists of 21837 clinical 12-lead ECGs, each lasting 10 seconds, from 18885 patients. The data set was collected and preprocessed as in the Physionet repository. We collected

for all experiments the ECG signals at a sampling rate of 100 Hz. For the three imputation and forecasting scenarios, we utilize 20% as target values. In all of the settings, the number of input channels is 12 as it is a 12-lead electrocardiogram. For the 248 time steps setting, the data set was preprocessed on crops, which corresponds to 69,764 training and 8,812 test samples.

Table 18: Electricity data set details.

| Description | Value |
| --- | --- |
| Train size | 817 |
| Test size | 921 |
| Training batch | 43 |
| Sample length | 100 |
| Data set features | 370 |
| Sample features | 37 |
| Conditional values | 90, 70, 50 |
| Target values | 10, 30, 50 |

**Electricity data set**  Table 18 contains details on the electricity data set, which was used for RM imputation. The electricity data set from the UCI repository (Dua & Graff, 2017) contains electricity usage data (in kWh) gathered from 370 clients which represent 370 features every 15 minutes. The data set was collected and preprocessed as in (Du et al., 2022). As the data set not contain missing values, we collected the complete data set and in our experiments and randomly dropped the values for the computation of targets according to the RM scenario. The data is already normalized and we present results in this setting. The first 10 months of data (2011/01 - 2011/10) are the test set, the following 10 months of data (2011/11 - 2012/08) the validation set and the left (2012/09 - 2014/12) the training set. We directly utilize the training and test set leaving the validation set out. We consider this a challenging task due to the fact that the test set contains many clients that were not present in the training set. The data set contains 817 samples of a length of 100 time steps with the 370 mentioned features. However, we observed faster convergence when applying feature sampling, specifically, splitting the 370 channels into 10 batches of 37 features each, then, passing to the network mini-batches of 43 samples each with 37 features and its respective length of 100 to ensure that we do not drop any data during training.

Table 19: MuJoCo data set details.

| Description | Value |
| --- | --- |
| Train size | 8000 |
| Test size | 2000 |
| Training batch | 50 |
| Sample length | 100 |
| Data set features | 14 |
| Conditional values | 50, 30, 20, 10 |
| Target values | 50, 70, 80, 90 |

**MuJoCo data set**  Table 19 contains details of the MuJoCo data set, which is a data set for physical simulation, created by the authors at (Rubanova et al., 2019) using the "Hopper" model from the Deepmind Control Suite. The hopper's initial placements and speeds are randomly sampled in such a way that the hopper rotates in the air before crashing to the earth. There are 10,000 sequences of 100 regularly sampled time points for each trajectory in the 14-dimensional data set. By convention, there is a 80/20 random split for training and testing. Both data sets were already preprocessed by the NRTSI authors in (Shan et al., 2021) to ensure a fair comparison.

**PEMS-BAY data set**  Table 20 contains details on the PEMS-Bay data set (Huang et al., 2019), which was compiled for The Performance Measurement System (PeMS) of the California Transportation Agencies

Table 20: PEMS-Bay data set details.

| Description | Value |
|---|---|
| Train size | 1200 |
| Test size | 50 |
| Training batch | 40 |
| Sample length | 200 |
| Data set features | 325 |
| Sample features | 65 |
| Conditional values | 150 |
| Target values | 50 |

(CalTrans). It represents a network of 325 traffic sensors in the California Bay Area. It contains traffic readings every five minutes for six months, from January 1st 2017 to May 31st 2017. The data set was collected from Cini et al. (2022). The methodology of Cini et al. (2022) required the creation of an adjacency matrix from the original data set due to the use of a graph neural network. We work directly with the data set, which has 52,116 time steps and 325 features. However, in order to obtain samples for training, we considered 200 time steps over the first 50,000 to get 250 samples, we set the first 240 for training and the 10 rest for testing. Then, we fit a standard scaler on the training set and transform both the training and the test sets. Finally, we feature sampled to obtain batches by a factor of 5, where for training set we obtained 5 batches of 240 samples with 200 time steps and 65 channels each. Finally, we iterate over the second dimension to pass batches of 40 for training.

Table 21: METR-LA data set details.

| Description | Value |
|---|---|
| Train size | 750 |
| Test size | 100 |
| Training batch | 50 |
| Sample length | 200 |
| Data set features | 207 |
| Sample features | 40 |
| Conditional values | 150 |
| Target values | 50 |

**METR-LA data set**   Table 21 contains details on the METR-LA data set, which represents the traffic data from a road network consisting of 207 loop detectors in a period between the 1st of March 2012 to the 30th of June 2012. Similar to the preprocessing of the PEMS-Bay data set, we selected from the original set the first 34,000 time steps and the first 200 features, we obtained 170 samples of 200 time steps each. We used the first 150 as training set and the remaining 20 for testing. Finally, we sampled features by a factor of 5, obtaining for training 5 batches of 150 samples with 200 time steps and 40 features each. For training we iterate over the first dimension to obtain batches of 50 samples.

**Solar data set**   Table 22 presents details on the solar data set. The original data set contains the solar power production records in the year 2006, sampled every 10 minutes from 137 photovoltaic power plants in Alabama State (Lai et al., 2018). However, as a conventional benchmark, we collected the data set from GluonTS (Alexandrov et al., 2020) which represents hourly sampled data. The task is to condition on 168 time steps to forecast the following 24. The whole data set contains 73 samples of 192 time steps and 128 features each in a chronological manner, we use the first 65 as training set and the remaining 8 as the set. Similar to the preprocessing applied to other data sets described above, we feature sample this data set by a factor of 2, where for the training set, for example, we obtained 2 batches of 65 samples with 192 time steps and 64 features each. We standard scale the train and test sets for training using training set statistics.

Table 22: Solar data set details.

| Description | Value |
|---|---|
| Train size | 130 |
| Test size | 16 |
| Training batch | 65 |
| Sample length | 192 |
| Data set features | 128 |
| Sample features | 64 |
| Conditional values | 168 |
| Target values | 24 |

Table 23: ETTm1 data set details.

| Description | Value |
|---|---|
| Train size | 33,865, 34,417, 34,000, 33,600, 33,200 |
| Test size | 11,490, 10,000, 11,420, 10,000, 10,000 |
| Training batch | 65, 127, 17, 14, 4 |
| Sample length | 120, 96, 480, 576, 1,052 |
| Data set features | 7 |
| Conditional values | 96, 48, 384, 288, 384 |
| Target values | 24, 48, 96, 288, 672 |

**ETTm1 data set**  Table 23 contains the ETTm1 data set details. This data set was created to investigate the amount of detail required for long-time series forecasting based on the Electricity Transformer Temperature (ETT), which is an important measure for the long-term deployment of electric power. The data set contains information from a compilation of 2-year data from two distinct Chinese counties. Here, we work with ETTm1 which covers data at a 15-minute level. The data is composed of the target value oil temperature and six power load features. We collected and preprocessed the data directly from Zhou et al. (2021), we forecast in each of the benchmarking horizons, utilizing train and test sets, coming from the original split of train/val/test set, which was for 12/4/4 months, respectively. As seen in Table 23, there are five different preprocessing settings implemented with regard to the forecasting horizon, the first table contains information on three and the second for the remaining two, where the main difference is the number of values used to condition on, and the targets to forecast. Similarly, there are some differences with respect to the batch size used during training. As the samples get longer, we utilize smaller batch sizes. Finally, for samples generation, as the test sets are very large with more than 10,000 samples each, we subset the test set in order to obtain batches.

# E  Additional technical information

## E.1  S4 Model

$$
A_{nk} = - \begin{cases} (2n+1)^1/2(2k+1)1/2 & \text{if } n > k \\ n+1 & \text{if } n = k \\ 0 & \text{if } n < k \end{cases} \tag{5}
$$

S4 is a deep structured state space model, which is built with four ideal components for time series analysis, specifically, for the handling of long sequences. First of all, it contains the continuous representation of SSMs previously introduced in Eq. 4, for which with the help of HiPPO matrices (Gu et al., 2020), it provides an importance score for each past time step through an online compression of signals and a novel and robust updating system (HiPPO-LegS) that scales for long sequences. In a nutshell, HiPPO aims to overcome the

issue of time series of different lengths and the vanishing gradient problem. HiPPO is referenced in 5, which compresses the scaled Legendre measure (LegS) operator for a uniform system update on the SMM vector $A$.

$$
\begin{aligned}
\overline{A} &= (I - \Delta/2 \cdot A)^{-1}(I + \Delta/2 \cdot A) \\
\overline{B} &= (I - \Delta/2 \cdot A)^{-1}\Delta B \\
\overline{C} &= C
\end{aligned}
\tag{6}
$$

Additionally, the HiPPO architecture provides the advantage of being able to handle irregularly sampled data with the help of a recurrent discretization procedure by a step size $\Delta$ using the bilinear method. The discretization procedure in Eq. 6 creates a sequence-to-sequence mapping that yields a recurrent state space; here, the SSMs can be computed like an RNN. Concretely, it can be viewed as a hidden state with a transition matrix $\overline{A}$.

$$
y_k = \overline{CAB}^k u_0 + \overline{CAB}^{k-1} u_1 + \ldots + \overline{CAB} u_{k-1} + \overline{CB} u_k \quad y = \overline{K} * u
\tag{7}
$$

$$
\overline{k} \in \mathbb{R}^L := K_L(\overline{A}, \overline{B}, \overline{C}) := (\overline{CAB}^i)_{i \in [L]} = (\overline{CB}, \overline{CAB}, \ldots, \overline{CAB}^{L-1})
\tag{8}
$$

The RNN described previously can be transformed into a CNN by unrolling in the discrete convolution step, which is shown in 7, where a single convolution can be computed efficiently with fast Fourier transform given the SSM convolution kernel $\overline{K}$ represented in Eq. 8. We strongly encourage the reader to refer to (Gu et al., 2022a) and (Gu et al., 2020) for further investigation of the S4 model and its components. From an external perspective, however, the S4 layer that is composed of multiple S4 blocks can be considered as a drop-in replacement for a one-dimensional (potentially dilated) convolutional layer, a RNN, or a transformer layer, which is particularly adapted to the requirements of time series analysis.

### E.2 Performance metrics

This section describes the performance metrics used throughout this work. We use a wide range of performance metrics for the computation of errors between targets and imputed values ($e_t = y - \hat{y}$), where for $y$ and $\hat{y}$ we apply the respective conditional masking for metrics computation ($m_{\text{eval}} = m_{\text{mvi}} \odot (1 - m_{\text{imp}})$).

$$
MAE = \frac{1}{n} \sum_{t=1}^{n} \sum_{k=1}^{K} |(y - \hat{y}) \odot m_{\text{eval}}|_{t,k}
\tag{9}
$$

$$
MSE = \frac{1}{n} \sum_{t=1}^{n} \sum_{k=1}^{K} ((y - \hat{y}) \odot m_{\text{eval}})^2_{t,k}
\tag{10}
$$

$$
RMSE = \sqrt{\frac{1}{n} \sum_{t=1}^{n} \sum_{k=1}^{K} ((y - \hat{y}) \odot m_{\text{eval}})^2_{t,k}}
\tag{11}
$$

$$
MRE = \frac{1}{n} \sum_{t=1}^{n} \sum_{k=1}^{K} m_{\text{eval}\,t,k} \frac{|(y - \hat{y})|_{t,k}}{y_{t,k}}
\tag{12}
$$

Firstly, from a deterministic perspective, specifically for the computation of absolute errors, we implemented mean absolute error (MAE) equation 9 which is determined by dividing the total absolute errors by the

total sample size of observations $n$. For the account of larger error sensitivity we implemented mean squared error (MSE) equation 10 which is the sum $\sum_{t=1}^{n}$ of squared errors $e_t^2$ divided by the total sample size of observations $n$, similarly, we implemented root mean squared error (RMSE) equation 11 to account for larger errors while preserving the error ranges in proportion to the observed mean, and to measure the precision of our imputations mean relative error (MRE) which as a percentage representation computes the mean of differences between the absolute errors $|y - \hat{y}|$ divided by their target values $y$.

### E.3 Training times

Table 27, Table 29 and Table 28 describe the number of training iterations and epochs used while training on each of the data sets.

Table 24: Imputations for RM, RBM, and BM scenarios on the PTB-XL data set.

| Model | MAE | RMSE |
|---|---|---|
| 20% RM on PTB-XL | | |
| Median | 0.1040 | 0.2071 |
| LAMC | 0.0678 | 0.1309 |
| DiffWave$D_0$ | 0.0047±2e-5 | 0.0175±3e-4 |
| DiffWave$D_1$ | 0.0043±4e-4 | 0.0177±4e-4 |
| CSDI | 0.0038±2e-6 | 0.0189±5e-5 |
| **CSDI$^{\mathbf{S4}}$** | **0.0031±1e-7** | 0.0171±6e-4 |
| SSSD$^{\text{SA}}D_0$ | 0.0052±3e-5 | 0.0229±4e-6 |
| SSSD$^{\text{SA}}D_1$ | 0.0045±3e-7 | 0.0181±4e-6 |
| SSSD$^{\text{S4}}D_0$ | 0.0044±2e-5 | 0.0137±1e-4 |
| **SSSD$^{\mathbf{S4}}\mathbf{D_1}$** | 0.0034±4e-6 | **0.0119±1e-4** |
| 20% RBM on PTB-XL | | |
| Median | 0.1074 | 0.2157 |
| LAMC | 0.0759 | 0.1498 |
| DiffWave$D_0$ | 0.0482±1e-3 | 0.1209±8e-3 |
| DiffWave$D_1$ | 0.0250±1e-3 | 0.0808±5e-3 |
| CSDI | 0.0186±1e-5 | 0.0435±2e-4 |
| CSDI$^{\text{S4}}$ | 0.0222±2e-5 | 0.0573±1e-3 |
| SSSD$^{\text{SA}}D_0$ | 0.0202±1e-3 | 0.0612±1e-2 |
| SSSD$^{\text{SA}}D_1$ | 0.0170±1e-4 | 0.0492±1e-2 |
| SSSD$^{\text{S4}}D_0$ | 0.0116±2e-4 | 0.0251±7e-4 |
| **SSSD$^{\mathbf{S4}}\mathbf{D_1}$** | **0.0103±3e-3** | **0.0226±9e-4** |
| 20% RM on PTB-XL | | |
| Median | 0.1252 | 0.2347 |
| LAMC | 0.0840 | 0.1171 |
| DiffWave$D_0$ | 0.0492±1e-3 | 0.1405±8e-3 |
| DiffWave$D_1$ | 0.0451±7e-4 | 0.1378±5e-3 |
| CSDI | 0.1054±4e-5 | 0.2254±7e-5 |
| CSDI$^{\text{S4}}$ | 0.0792±2e-4 | 0.1879±1e-4 |
| SSSD$^{\text{SA}}D_0$ | 0.0493±1e-3 | 0.1192±7e-3 |
| SSSD$^{\text{SA}}D_1$ | 0.0435±3e-3 | 0.1167±1e-2 |
| SSSD$^{\text{S4}}D_0$ | 0.0415±1e-3 | 0.1073±5e-3 |
| **SSSD$^{\mathbf{S4}}\mathbf{D_1}$** | **0.0324±3e-3** | **0.0832±8e-3** |

Table 25: MAE, MSE, MRE for PEMS-BAY and METR-LA data sets. All baseline results were collected from Cini et al. (2022).

| Model | MAE | MSE | MRE |
|---|---|---|---|
| \multicolumn{4}{c}{25% RM on PEMS-Bay} | | | |
| Mean | 5.42±0.00 | 86.59±0.00 | 8.67±0.00 |
| KNN | 4.30±0.00 | 49.80±0.00 | 6.88±0.00 |
| MF | 3.29±0.01 | 51.39±0.64 | 5.27±0.02 |
| MICE | 3.09±0.02 | 31.43±0.41 | 4.95±0.02 |
| VAR | 1.30±0.00 | 6.52±0.01 | 2.07±0.01 |
| rGAIN | 1.88±0.02 | 10.37±0.20 | 3.01±0.04 |
| BRITS | 1.47±0.00 | 7.94±0.03 | 2.36±0.00 |
| MPGRU | 1.11±0.00 | 7.59±0.02 | 1.77±0.00 |
| **GRIN** | **0.67±0.00** | **1.55±0.01** | **1.08±0.00%** |
| SSSD$^{S4}$ | 0.97±0.01 | 2.98±0.03 | 1.42±0.01 |
| \multicolumn{4}{c}{25% RM on PEMS-Bay} | | | |
| Mean | 7.56±0.00 | 142.22±0.00 | 13.10±0.00 |
| KNN | 7.88±0.00 | 129.29±0.00 | 13.65±0.00 |
| MF | 5.56±0.03 | 113.46±1.08 | 9.62±0.05 |
| MICE | 4.42±0.07 | 55.07±1.46 | 7.65±0.12 |
| VAR | 2.69±0.00 | 21.10±0.02 | 4.66±0.00 |
| rGAIN | 2.83±0.01 | 20.03±0.09 | 4.91±0.01 |
| BRITS | 2.34±0.00 | 16.46±0.05 | 4.05±0.00 |
| MPGRU | 2.44±0.00 | 22.17±0.03 | 4.22±0.00 |
| **GRIN** | **1.91±0.00** | **10.41±0.03** | **3.30±0.00** |
| SSSD$^{S4}$ | 2.83±0.02 | 21.95±0.14 | 5.59±0.08 |

Table 26: Forecasting results for the PTB-XL data set.

| Model | MAE | RMSE |
|---|---|---|
| Median | 0.134 | 0.273 |
| CSDI | 0.165±0.0009 | 0.302±0.0004 |
| CSDI$^{S4}$ | 0.120±0.0002 | 0.246±0.0001 |
| SSSD$^{SA}$ | **0.087±0.008** | 0.220±0.012 |
| SSSD$^{S4}$ | 0.090±0.003 | **0.219±0.006** |

Table 27: Training on the proposed PTB-XL data sets. training epochs (e) and iterations (i) on the proposed PTB-XL data set.

| Model | PTB 248 | PTB 1000 |
|---|---|---|
| CSDI | 200 (e) | 200 (e) |
| CSDI$^{S4}$ | 200 (e) | 200 (e) |
| DiffWave | 150,000 (i) | 150,000 (i) |
| SSSD$^{SA}$ | 150,000 (i) | 150,000 (i) |
| SSSD$^{S4}$ | 150,000 (i) | 150,000 (i) |

Table 28: CSDI training on benchmarking data sets. contains the CSDI training epochs (e) on the benchmarking data sets.

| Dataset | Setting | Epochs |
|---|---|---|
| Electricity | Imputation 10% | 200 (e) |
| Electricity | Imputation 30% | 200 (e) |
| Electricity | Imputation 50% | 200 (e) |
| MuJoCo | Imputation 70% | 200 (e) |
| MuJoCo | Imputation 80% | 200 (e) |
| MuJoCo | Imputation 90% | 200 (e) |
| ETTm1 | Forecast 24 | 200 (e) |
| ETTm1 | Forecast 48 | 200 (e) |
| ETTm1 | Forecast 96 | 200 (e) |
| ETTm1 | Forecast 288 | 200 (e) |
| ETTm1 | Forecast 672 | 200 (e) |

Table 29: SSSD$^{S4}$ training on benchmarking data sets. contains the SSSD$^{S4}$ training iterations (i) on the benchmarking data sets across diverse baselines.

| data set | Setting | Iterations |
|---|---|---|
| Electricity | Imputation 10% | 150,000 (i) |
| Electricity | Imputation 30% | 150,000 (i) |
| Electricity | Imputation 50% | 150,000 (i) |
| PEMS-BAY | Imputation 25% | 350,000 (i) |
| METR-LA | Imputation 25% | 250,000 (i) |
| MuJoCo | Imputation 70% | 232,000 (i) |
| MuJoCo | Imputation 80% | 160,000 (i) |
| MuJoCo | Imputation 90% | 150,000 (i) |
| Solar | Forecast 24 | 100,000 (i) |
| ETTm1 | Forecast 24 | 212,000 (i) |
| ETTm1 | Forecast 48 | 150,000 (i) |
| ETTm1 | Forecast 96 | 250,000 (i) |
| ETTm1 | Forecast 288 | 250,000 (i) |
| ETTm1 | Forecast 672 | 250,000 (i) |

## F   Multimedia Appendix

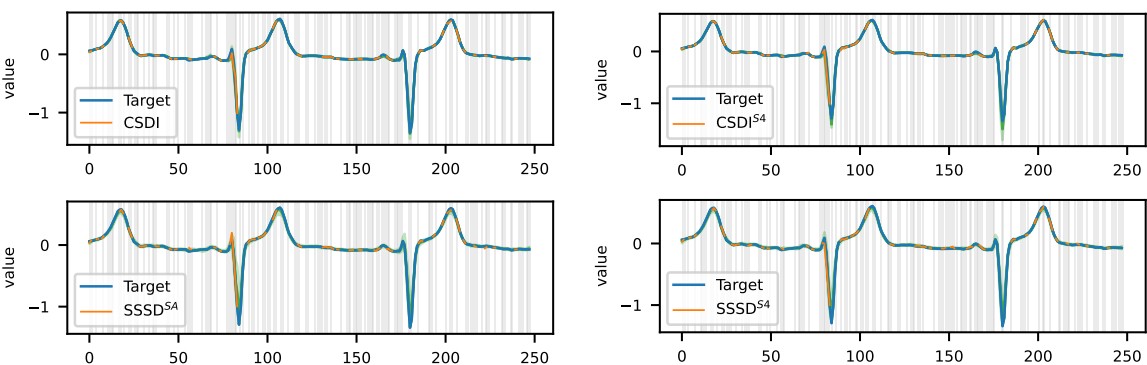

Figure 6: PTB-XL RM imputations of a V3 lead for an ECG from a patient with an anterior myocardial infarction (AMI). The figure shows 100 RM imputations on a single sample tested at 50% for four imputers on the PTB-XL data set. As clearly seen from the plots above, all the diffusion models are highly capable of reconstructing time series, even when there is a high ratio of missing values in a RM scenario. Quantitatively, as seen in the main paper, we observed that CSDI$^{S4}$ outperforms the rest of the models on this data set. The differences in terms of imputation quality can hardly be observed visually as the rest of the models present slight green shaded areas surrounding the orange imputed areas which represent the imputation quantiles.

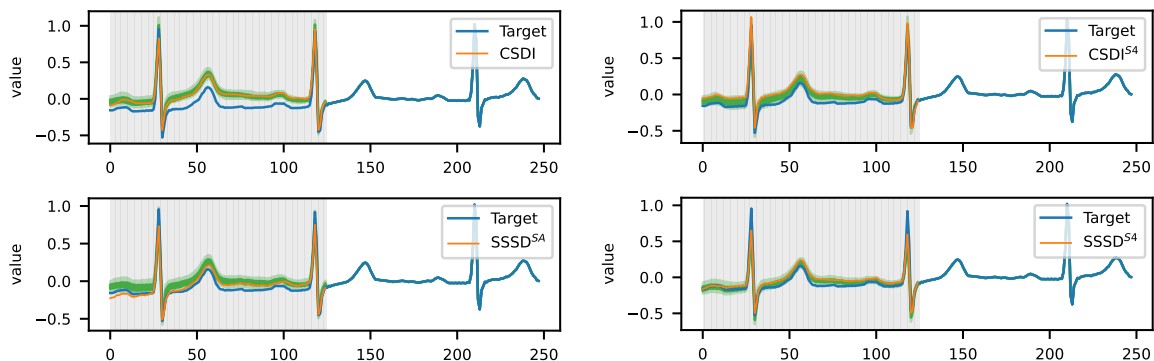

Figure 7: PTB-XL RBM imputations of a V4 lead for an ECG from a patient with an AV block (AVB). The figure shows 100 RBM imputations on a single sample tested at 50% for four imputers on the PTB-XL data set. As previously discussed, RBM assumes that the missing blocks are located at different time steps on each feature. Technically, in this setting, the diffusion models are capable of inferring the missing segments in a given channel from neighboring channels at the same time steps, which empirically seems to be a factor that enables an overall very good reconstruction across all models.

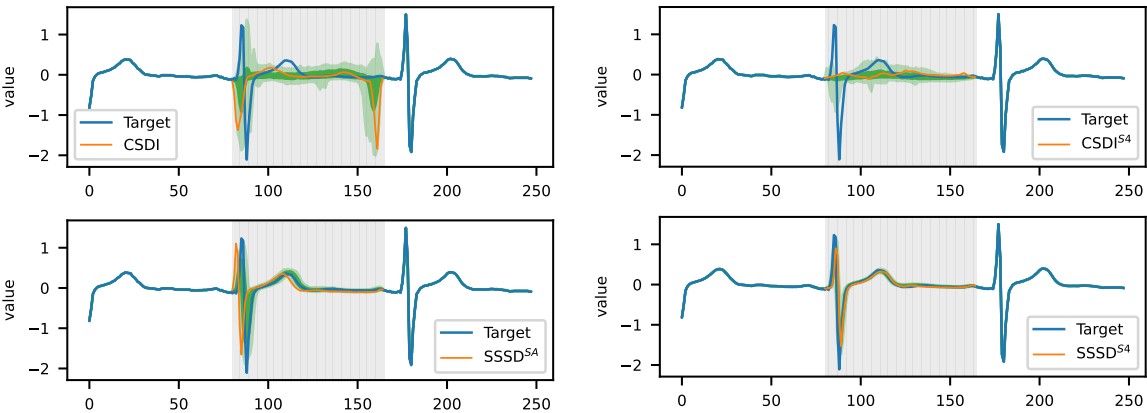

Figure 8: PTB-XL BM imputations for lead V2 for an ECG collected from a healthy patient. The figure shows four BM imputations tested at 30% on the PTB-XL data set reiterating the qualitative differences in imputation quality already demonstrated in the main text.

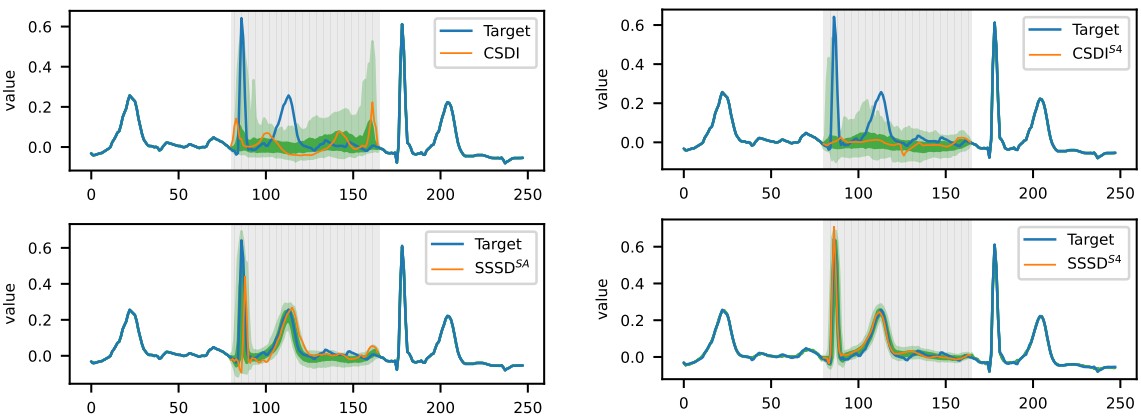

Figure 9: PTB-XL BM imputations for lead V6 for a normal heart condition. The figure shows four BM imputations tested at 30% on the PTB-XL data set.

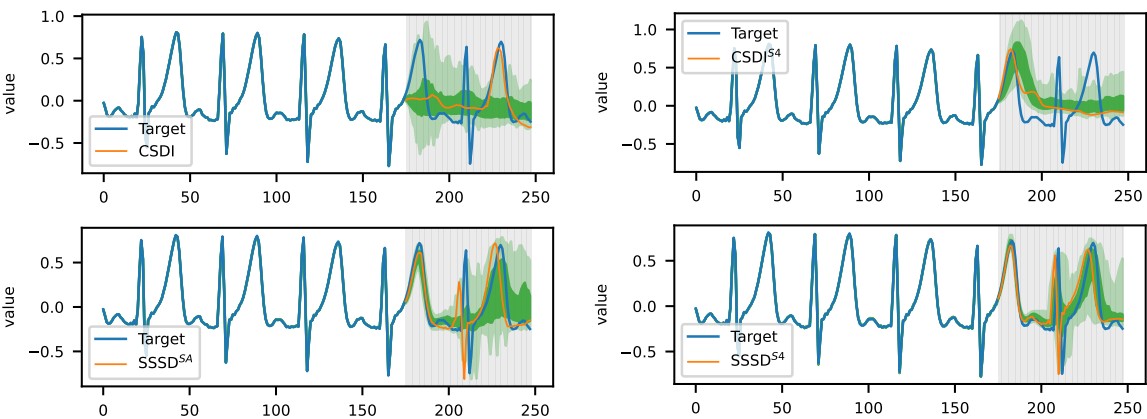

Figure 10: PTB-XL TF for lead V3 for an ECG from a patient with left ventricular hypertrophy. The figure shows 100 BM imputations on a single sample tested at 50% for four imputers on the PTB-XL data set. CSDI model is overall learning the trend that the series has, with a few correct feature generations as seen in the plot, nevertheless, we observe that the imputation falls outside the 0.05 and 0.95 quantile range, which is basically an outlier. CSDI$^{S4}$ improves the quality of the generations, as its quantiles are less diverse and start to follow the signals patterns, however, it seems that after a certain number of steps, the learning decrease dramatically. On the contrary, SSSD$^{SA}$ and SSSD$^{S4}$ correctly capture the characteristics of the signal. In particular, SSSD$^{S4}$ maintains a tight interquartile range even at longer forecasting horizons.

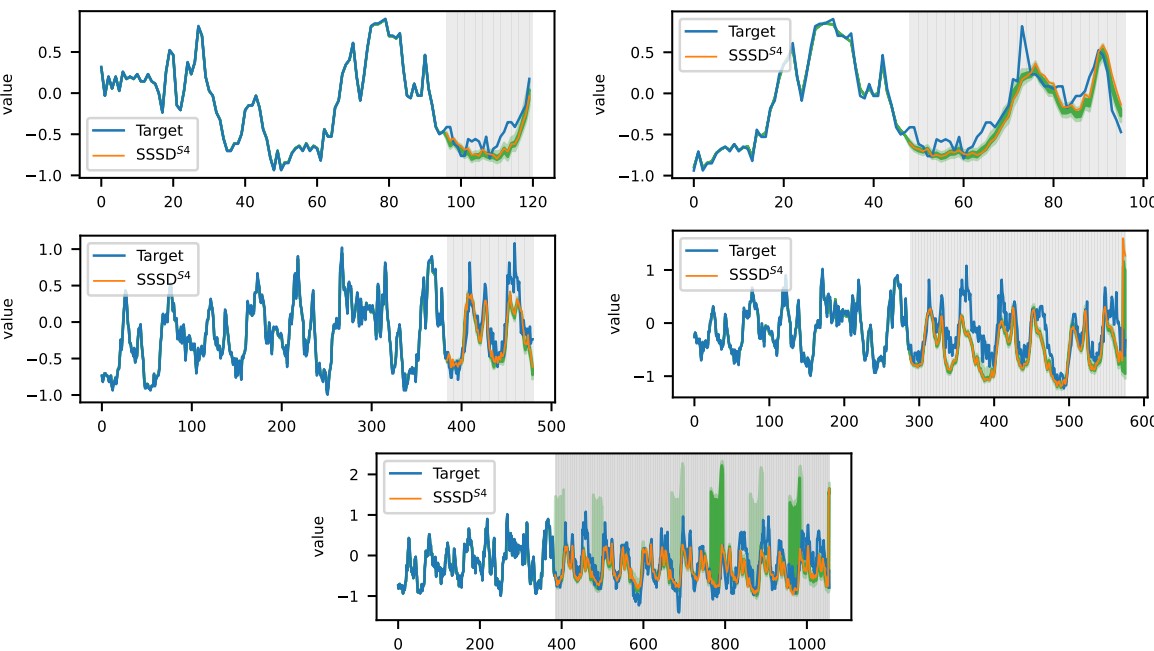

Figure 11: ETTm1 TF for the five different forecasting settings. From top to bottom 24, 48, 96, 288, 384 forecasting horizon targets. These plots display the complete sample including the conditional part.

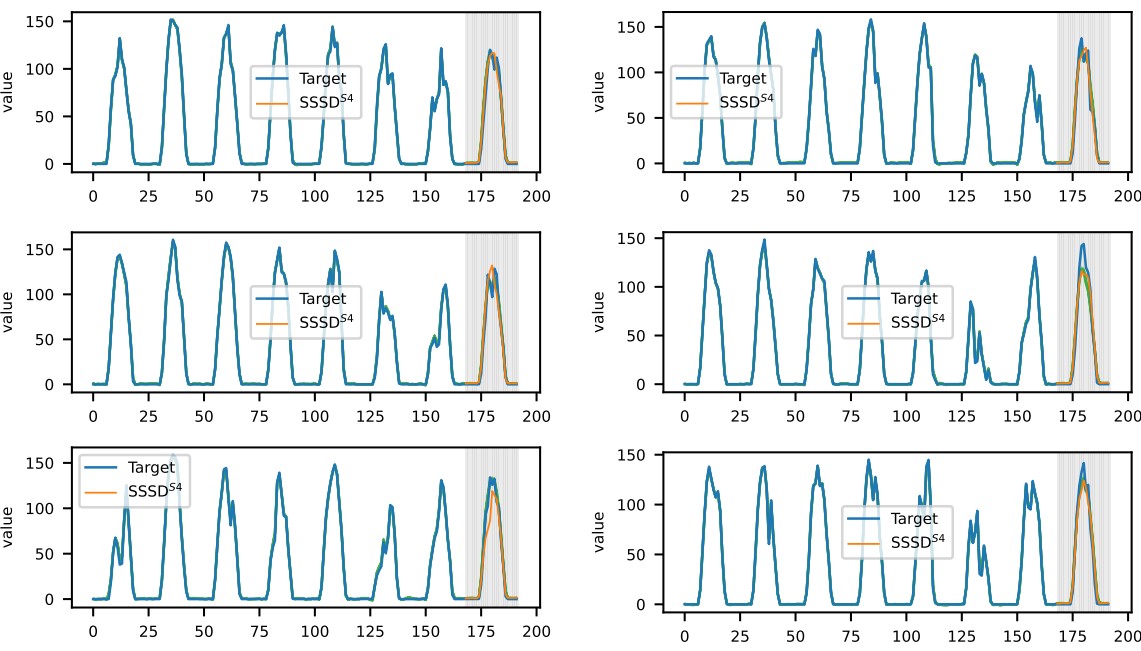

Figure 12: TF for six different channels of the Solar data set. These plots display the complete sample including the conditional part, see Figure 13 for more detailed plots of the imputed area only.

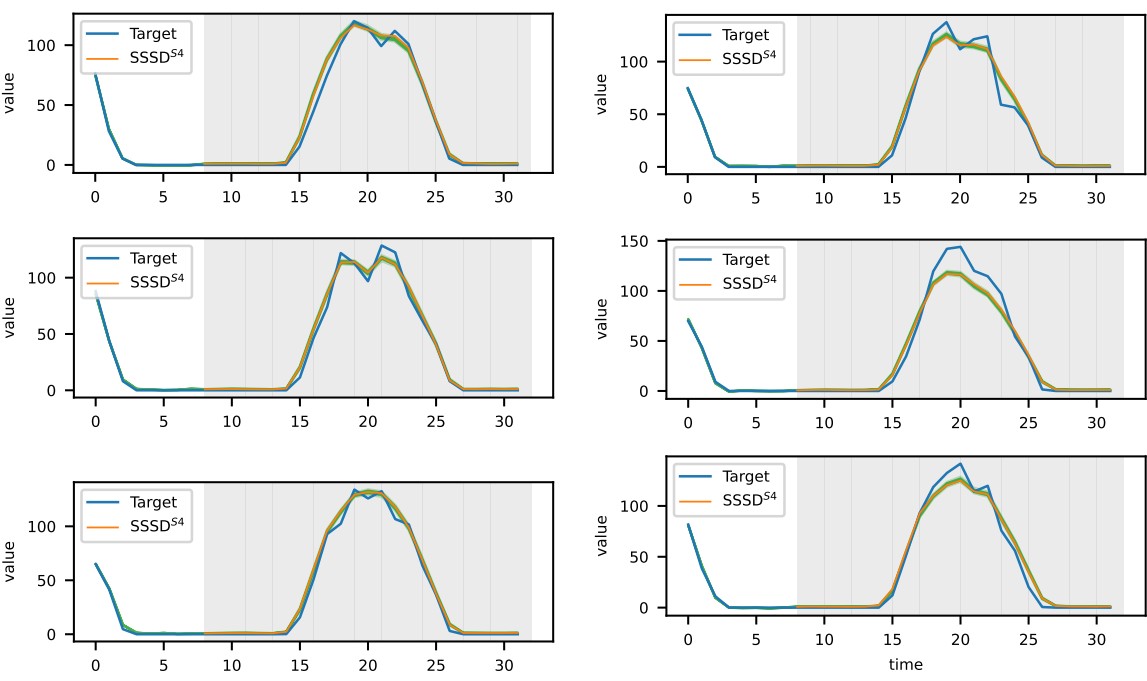

Figure 13: TF for six different channels of the Solar data set. These plots display only the imputed area.

