# OpenReview forum: "Diffusion-based Time Series Imputation and Forecasting with Structured State Space Models"
_TMLR — Accepted by TMLR_

### Review · Reviewer_5Tb4 · 2022-12-19

**Summary Of Contributions:**

The authors propose to combine the DiffWave diffusion model for time series with structured state-space models (S4) as internal layers to yield a conditional diffusion model that can be used for time series imputation. They show that this outperforms many existing time series imputation baselines on different real-world datasets.

**Audience:**

Yes

**Broader Impact Concerns:**

There are no boarder impact concerns.

**Claims And Evidence:**

Yes

**Requested Changes:**

As mentioned above, the relation of the studied missingness settings to the known types from the literature (MCAR, MAR, MNAR) should be made more clear. Also, the strong baselines, such as the Autoformer, should be compared against in all experiments where they can plausibly be used. Besides, there are a few typos and minor issues that should be fixed.

**Strengths And Weaknesses:**

Strengths:
- Time series imputation is an important problem.
- The proposed model seems to exhibit strong empirical performance on many tasks.

Weaknesses:
- The combination of the two existing DiffWave and S4 models seems slightly incremental, but that should normally not be a problem for TMLR.
- The selection of baselines in the experiments is sometimes a bit unclearly motivated.

Major comments:
- In the imputation literature, the different types of missingness are usually called missing completely at random (MCAR), at random (MAR), and not at random (MNAR) [1]. It seems like what you call RM is MCAR, while MNR, BM, and TF are all types of MAR. MNAR would imply that the missingness actually correlates with the underlying true value, which I don't think you consider. Especially your MNR nomenclature is therefore a bit confusing and it would probably be better to use the standard terms or at least clearly explain their correspondence to your settings.
- There are many baselines that seem pretty strong, but only show up in selected experiments, such as NAOMI, NRTSI, SAITS, and the Autoformer. It is a bit unclear to the reader how these sets of baselines are selected and why these strong baselines would not be used in the other experiments. Could for instance the Autoformer not also be used on the solar data, or electricity, or ECG? I would generally expect at least the strongest baselines to be used in all experiments where they can plausibly be applied.

Minor comments:
- In the figures, why is always a random sample plotted instead of the mean of the imputation distribution? Generally, I would find the mean more informative, since the quality between samples can presumably vary a lot.
- In Sec. 2, should it say $m_{imp} \in \lbrace 0,1 \rbrace^{L \times K}$ or can the indicators indeed take values in the interval between 0 and 1, as written in the paper?
- Please use \citep{} for parenthetical citations.
- Could you discuss the tradeoff of mapping the input channels to the diffusion dimension in a bit more detail? Especially to readers that are not perfectly familiar with the related literature, this could help to assess what the implied design decisions entail.
- Fig. 2: Tahn -> Tanh
- Fig. 2: $\sigma2$ -> $\sigma^2$
- Tab. 2: maybe you could just write in the caption that all values are on the order 1e-3, to remove clutter from the table
- Tab. 4: The error bars seem quite large. Is the proposed method's performance really 0.027 +/- 4.32 or should it be 0.027 +/- 0.0432 or something? If the error bars are really that large, shouldn't all methods be boldened in the table, because their error bars would overlap? Does this imply anything about the robustness of the proposed method?

[1] Donald B. Rubin, Inference and missing data, Biometrika, Volume 63, Issue 3, December 1976, Pages 581–592, https://doi.org/10.1093/biomet/63.3.581

---

### Review · Reviewer_kCUh · 2022-12-24

**Summary Of Contributions:**

The authors present SSSD (Structured State Space Diffusion), a model which combines recent innovations in sequence modeling (i.e. S4 layers) with conditional diffusion models for time-series imputation. The main methodological contributions are as follows:

1) The authors propose to augment the DiffWave model architecture by introducing two S4 layers in each residual block in the UNet-based denoiser (Fig. 2 in the manuscript) to enhance its capability for time series modeling.
2) Diffusion is either applied on the entire input signal ($D_0$) or in the regions of the input which need to be imputed only ($D_1$). The choice of $D_1$ is justified using empirical results.
3) The conditioning signal to the reverse process is the concatenation of the imputation masks and the imputed signal.

The authors justify the choice of S4 layers using a small ablation experiment while also presenting empirical results on various benchmarks (like PTB-XL, MuJoCo etc.) in tasks across settings like RM, MNR, BM and Forecasting (which is a special case of BM-missingness).

**Audience:**

Yes

**Broader Impact Concerns:**

The authors present a generative model in this work, so Broader concerns regarding generation of fake data apply to this method as well.

**Claims And Evidence:**

Yes

**Requested Changes:**

**Requested Changes (Major):**

1) The authors have provided some ablation results related to S4 layers in the Appendix. However, it appears that these results are based on the 20% BM scenario on the PTB-XL dataset. Given that, the introduction of S4 layers in the UNet model is one of the primary contributions of this work, I would expect these ablations across multiple imputation settings (including RM, MNR) and missingness ratios (20%, 30% etc.). It would suffice to show these results for at-least one dataset (maybe PTB-XL?).

2) Diffusion and Training Hyperparameters: It would help to observe the impact of using larger number of timesteps during training and inference on the imputation performance. The authors do mention about this briefly in Appendix A.1 as: "We found that the fewer diffusion steps, the
faster the network converges during training, however, at the cost of less accurate results". It would be great if authors can present some quantitative results about the same on a dataset (maybe PTB-XL) since the Inference speed vs performance tradeoff is central to diffusion models.

3) Did the authors experiment with a setting where there is a mismatch between the missingness ratios during training and inference? For instance, a setting when the model is trained at 20% RM but is evaluated at 30% RM (and vice-versa)? Another possibility could be training for one missingness scenario (maybe 20% RM) and evaluating for MNR or BM? From my understanding of Section 4.1, the authors always train and evaluate across the same setting. It would be interesting to benchmark the generalization capabilities of the imputation model.

4) For the PTB-XL dataset results in Section 4.2, the authors consider a 20% missingness ratio across all settings (RM, MNR and BM) (Table 1 and 20). However, for other datasets (like MuJoCo (Table 2), Electricity (Table 3 and 21)) the authors consider multiple sparsity ratios for only the RM scenario. Is there a specific reason for the selection of specific settings for specific datasets? If possible, I would strongly recommend the authors to present comparisons across multiple imputation ratios (20%, 30% etc.) across multiple settings for atleast one dataset (maybe PTB-XL or whichever the authors deem fit).

**Requested Changes (Minor)**

The following changes are mostly meant to improve the readability of the paper:

1) Consider adding a more (formal) explanation of different missingness scenarios in Section 2 which is consistent with existing survey literature [1,2]. For instance does the Random Missing (RM) scenario correspond to MCAR or MAR as described in [1]? Similarly for other missingness scenarios considered in the paper.

2) The authors mention about a channel splitting approach for datasets with more than 100 channels. However, I could not find an explanation of the same in the main text or the Appendix. If already discussed, can the authors point me to a relevant section in the paper. If not, I would strongly recommend the authors to explain the approach. Moreover, the authors hint that the method might not perform well on datasets with more than 100 channels per time point without modifying the approach with suitable approaches. I would encourage the authors to mention this as possible future directions to the proposed approach (among others).

3) In Section 2 (Diffusion models), consider citing [3,4] since Score matching with gaussian perturbation kernels $p(x_t|x_0)$ is equivalent to the “epsilon”-prediction framework in SoTA Diffusion models (as also shown in Ho et al.). Moreover, for Video Diffusion models consider citing [5] which is one of the first papers to propose diffusion for Video prediction. Lastly, for Conditional Diffusion models, consider citing [6] which discusses different conditioning mechanisms for forward and reverse processes in the Diffusion backbone. It is worth noting, that list is in no way exhaustive, and the authors might want to look into relevant literature.

4) There are several instances of typos and missing references in the paper. For example, references to relevant tables in Sections 4.2 and 4.3 on the ECG data are missing. In Section 4.3, “table 4” should be “Figure 4”. I would highly recommend the authors to fix these (but not limited to) minor issues in subsequent revisions.

References:

[1] A survey on data imputation techniques: Water distribution system as a use case

[2] Missing value imputation: a review and analysis of the literature (2006–2017)

[3] Generative Modeling by Estimating Gradients of the Data Distribution

[4] Score-Based Generative Modeling through Stochastic Differential Equations

[5] Diffusion Probabilistic Modeling for Video Generation

[6] DiffuseVAE: Efficient, Controllable and High-Fidelity Generation from Low-Dimensional Latents

**Strengths And Weaknesses:**

**Strong Aspects:**

1) The motivation behind the proposed methodology is clear. The introduction of S4 layers adapted in the UNet architecture to handle imputation scenarios in time-series data is novel and makes sense. The authors also provide a nice ablation experiment in the Appendix which justifies the use of S4 layers as architectural components.

2) The methodological contributions are supported by sound empirical results (both qualitatively and quantitatively). The proposed method seems to perform better (or at par) than competing SoTA baselines over multiple benchmarks across missingness settings like RM, MNR, BM, or Forecasting. Moreover, the quantitative results have been discussed in detail in the main text while most of the training details for SSSD and other baselines have been covered in significant detail in the Appendix.

3) With relevant adjustments, the authors show that the proposed method can also adapt to datasets with more than 100 channels on the Electricity dataset with significant improvements over other baselines which is interesting!

**Weaknesses**: Given the strong aspects, I do have some concerns/questions about the experiments in general, which are discussed in more detail in the "Requested Changes" section.

---

### Review · Reviewer_vTws · 2023-01-05

**Summary Of Contributions:**

The paper proposes a new hybrid model for time series imputation -- which combines a structured state-space models together with deep learning components. This allows the model to pick up longer-term dependencies more easily, while still learning complex patterns from data.

**Audience:**

Yes

**Broader Impact Concerns:**

-

**Claims And Evidence:**

Yes

**Requested Changes:**

I think a much more detailed description of the model architecture and training would be crucial to include -- along with concrete answers to the questions above.

**Strengths And Weaknesses:**

Strengths
---
1. As many architectures are only compatible with regular discretely sampled data, data imputation techniques are important to allow for common deep neural networks to be trained on data with missingness.

2. The strong experimental results provide strong support for the benefits of using SSSD for data imputation, despite lower performance in the forecasting scenario.

Weaknesses
---
1. The paper is mostly well written, but I did find it a little difficult to understand how the model was concretely implemented. In particular, 1) how the model utilizes state space components, 2) what loss function and batching procedure was used in training, and 3) how imputation is performed at run-time. It would be great to get all architectural components laid out in equation form for detailed analysis.

2. While cited literature on imputation models is comprehensive, there is a lot of prior work on hybrid deep learning models that incorporate state space components which should be referenced – e.g. Kalman VAE, Deep Kalman Filters, Deep State Space Models, Deep Factors for forecasting etc to name a few.

3. Are all models compatible with backward imputation, and does the SSSD use backward and forward smoothing in all imputation tasks?

4. Could the authors describe imputation applications where a backward step might be useful? Where imputation as a pre-cursor to train other models, would there be any concerns about leaking future information into the imputed data if a backward smoothing step is used?

5. For consistency, all models should use a common set of benchmarks and metrics for better comparisons.

---

### Author Response · Authors · 2023-05-02
**Update**

Abdul Fatir thankfully reported a potential issue due to data leakage between train and test set while preprocessing data from the Solar dataset. We quickly confirmed that this was actually the case. After fixing the issue, we trained a model from scratch on this split, which lead to a test MSE of 5.03e2 ± 1.06e1 (as compared to 2.7e2 ± 0.43e1). The result stays on top of the baselines reported in Table 4. We apologize for this shortcoming and updated the preprint version of the article as well as the preprocessing code in the official code repository.

---

### Decision · Action_Editors · 2023-02-21

**Recommendation:** Accept as is

**Comment:**

The authors propose a conditional diffusion model for time series imputation. Their model relies on combining DiffWave and structured state-space models. They argue that this approach addresses some of the shortcomings of existing models and can handle long-term dependencies in time series data. The authors stress the importance of using probabilistic imputation methods that provide samples of different plausible imputations instead of a single imputation.

Time series imputation is an important practical problem, e.g., in healthcare data analysis. Strengths of the paper include systematic ablations of their approach, as well as good practical performance, especially in high-dimensional time series.

The authors successfully resolved minor issues relating to the implementation (e.g., uni- vs. bidirectional context), their used nomenclature regarding different missingness patterns, added a consistent baseline method, and improved the paper structure for better readability.

Overall, I recommend the paper for acceptance.

**Audience:**

Yes, there is a substantial audience who may be interested in applications of diffusion models for imputation tasks.

**Claims And Evidence:**

No concerns.